# Effective combination of arugula vermicompost, chitin and inhibitory bacteria for suppression of the root-knot nematode *Meloidogyne javanica* and explanation of their beneficial properties based on microbial analysis

Mahsa Rostami[1], Akbar Karegar[ID][1]*, S. Mohsen Taghavi[1], Reza Ghasemi-Fasaei[2], Abozar Ghorbani[3¤]

1 Department of Plant Protection, School of Agriculture, Shiraz University, Shiraz, Iran, 2 Department of Soil Science, School of Agriculture, Shiraz University, Shiraz, Iran, 3 Plant Virology Research Centre, School of Agriculture, Shiraz University, Shiraz, Iran

¤ Current address: Nuclear Agriculture Research School, Nuclear Science and Technology Research Institute (NSTRI), Atomic Energy Organization of Iran (AEOI), Karaj, Iran
* karegar@shirazu.ac.ir

## Abstract

Root-knot nematodes (*Meloidogyne* spp.) are dangerous parasites of many crops worldwide. The threat of chemical nematicides has led to increasing interest in studying the inhibitory effects of organic amendments and bacteria on plant-parasitic nematodes, but their combination has been less studied. One laboratory and four glasshouse experiments were conducted to study the effect on *M. javanica* of animal manure, common vermicompost, shrimp shells, chitosan, compost and vermicompost from castor bean, chinaberry and arugula, and the combination of arugula vermicompost with some bacteria, isolated from vermicompost or earthworms. The extract of arugula compost and vermicompost, common vermicompost and composts from castor bean and chinaberry reduced nematode egg hatch by 12–32% and caused 13–40% mortality of second-stage juveniles in vitro. Soil amendments with the combination vermicompost of arugula + *Pseudomonas. resinovorans* + *Sphingobacterium daejeonense* + chitosan significantly increased the yield of infected tomato plants and reduced nematode reproduction factor by 63.1–76.6%. Comparison of chemical properties showed that arugula vermicompost had lower pH, EC, and C/N ratio than arugula compost. Metagenomics analysis showed that *Bacillus*, *Geodermatophilus*, *Thermomonas*, *Lewinella*, *Pseudolabrys* and *Erythrobacter* were the major bacterial genera in the vermicompost of arugula. Metagenomics analysis confirmed the presence of chitinolytic, detoxifying and PGPR bacteria in the vermicompost of arugula. The combination of arugula vermicompost + chitosan + *P. resinovorans* + *S. daejeonense* could be an environmentally friendly approach to control *M. javanica*.

**Data Availability Statement:** All relevant data are within the paper and its Supporting Information files.

**Funding:** The authors gratefully acknowledge the financial support from Shiraz University. The funders had no role in study design, data collection and analysis, decision to publish, or preparation of the manuscript.

**Competing interests:** The authors declare no conflict of interest

## Introduction

Root-knot nematodes (*Meloidogyne* spp.) are among the most damaging agricultural pests and cause significant economic losses. They specifically attack the root vascular system and cause nutrient deficiencies in the host and disruption of water transport. Visible aboveground symptoms include stunted growth, wilting, chlorosis, and lower crop yields. These parasites have a remarkable ability to infect and multiply in the roots of numerous plant species, which can even lead to crop failure. Conventional approaches to nematode control, such as chemical nematicides, have proven effective to some degree, but are associated with serious environmental and health concerns. Many efforts have been made to direct the management of plant parasitic nematodes towards environmentally friendly methods [1,2]. Increasing awareness of the harmful effects of chemical pesticides on the environment, non-target organisms, and human health has increased the need for alternative, environmentally friendly, and sustainable solutions [3]. In this context, the use of organic amendments, beneficial microorganisms and natural compounds has emerged as a promising strategy for nematode control in agriculture [4]. Among these strategies, the use of organic soil amendments such as compost and vermicompost is an effective method for the control of plant parasitic nematodes that could improve soil quality and plant health [5].

Vermicompost is a soil amendment and a biological control agent against fungi and bacteria that can improve plant growth and resistance to agricultural pests [6]. The liquid extract of vermicompost (vermiwash), which contains proteins, enzymes, vitamins, hormones, bioavailable minerals and decomposing bacteria, can suppress plant pathogens and increase crop productivity [7]. The application of biochar and vermicompost has demonstrated its effectiveness in mitigating the stress caused by the rice root-knot nematode *M. graminicola*. In particular, the application of 1.2% biochar and 5% vermicompost has shown promising results in controlling the infestation of rice plants [8]. In vitro evaluations of vermicompost and its extracts on tomato root-knot nematodes also showed promising results. As the concentration of vermicompost extract increased, the hatching rate of eggs decreased and the mortality rate of *M. incognita* second-stage juveniles (J2s) increased significantly. In addition, application of vermicompost in pot experiments with tomato plants resulted in reduction in the number of root knots, indicating its potential as an effective nematode control agent [9]. Different types of vermicompost derived from different plant wastes have dissimilar effects on nematode-infected plants. For example, significant improvements in growth parameters of nematode-infected tomato plants in the inoculated soil were observed when vermicompost from sources such as Saw Dust + Cow dung and Taro Leaves + Cow dung were incorporated [10]. Moreover, the different forms of vermicompost application play an active role in regulating *Meloidogyne* infestation in plants. In one study, both liquid vermicompost applied by foliar spraying and root uptake and solid vermicompost applied by soil amendment showed a significant reduction in the average number of root-knots compared to the control. However, vermicompost taken up through root seemed to be more effective. When growth parameters were evaluated, treatments via root uptake and foliar application showed higher vegetative expression compared to the supplementation method [11]. Although some studies have shown positive results using vermicompost against nematodes, the exact mechanisms of action are not fully understood. This study includes a microbial analysis of vermicompost, which leads to a better understanding of the biocontrol process.

Worm excrement contains various microorganisms such as bacteria, fungi and protozoa. The prominent microbial community and the type of vermicompost substrate materials are factors that determine the properties of vermicompost for agricultural applications. Earthworm activity alters the bacterial community of vermicompost during different stages of

composting [6,12]. Vermicompost is rich in beneficial bacteria, including plant growth-promoting rhizobacteria (PGPR), which can suppress nematode infection. Knowledge of the biodiversity of vermicompost bacteria helps to understand their useful properties in agriculture [13]. It showed that *Bacillus safensis*, *B. megaterium*, *Pseudomonas resinovorans*, *Lysinibacillus* sp., *L. fusiformis* and *Sphingobacterium daejeonense*, isolated from vermicompost or earthworms, have antagonistic properties against *M. javanica* [14]. Metagenomics analyses of bacteria in the gut of two common earthworms (*Perionyx xcavates* and *Eisenia foetida*), which play a major role in the decomposition of vermicompost, showed that *Proteobacteria* and *Firmicutes* are the dominant bacterial phyla in vermicompost [15]. The role of *Proteobacteria* is nitrogen fixation in soil [16], while *Firmicutes* eliminate pathogenic microbes [17]. Comparison of microbial structure and quality of vermicompost with aerobic compost by microbiome analysis showed that the dominant fungi and bacteria were different. The vermicomposting process of organic material enriched the microbial community [18].

Another determining factor for the quality of vermicompost is the type of substrate used for earthworm activity. Different substrates have been studied to produce better vermicompost. Animal manure, kitchen waste, agricultural residues, industrial wastes and some plants are used to produce more effective vermicompost [7,19]. Chitin is a polysaccharide widely distributed in nature and is found in crustaceans, insects and fungi. The main source of industrial chitin is shrimp and crab shells [20]. Due to its physical, chemical and biological properties, it has the potential to enrich vermicompost. Chitin and its derivatives can control plant diseases and induce resistance in the host plant [21]. Enrichment of vermicompost from cattle manure with chitin improved its properties, increased the growth parameters of infected tomato plants and reduced the population of *M. incognita* in root and soil [22]. It was shown that the addition of chitin to soil reduced the population of *Meloidogyne incognita* in tomato roots [23].

Another substrate for vermicomposting is inhibitor plants, which can reduce the damage of plant pathogens. Plant derivatives are effective organic additives for nematode control [24]. Many plants such as castor bean (*Ricinus communis* L.), arugula (*Eruca sativa* Mill.) and chinaberry (*Melia azedarach* L.) have been shown to effectively reduce nematode reproduction and improve plant growth [25,26].

In most studies, the extract or powder of the inhibitory plants was used for nematode control. However, to our knowledge, there is no report on the nematicidal effect of their vermicompost or microbial structure. This study aimed to investigate the synergistic effect of arugula vermicompost, chitin and inhibitory bacteria, in suppressing *M. javanica* populations in the rhizosphere, and to understand the mechanisms underlying the nematode suppressive effect of the best compost against root-knot through a comprehensive microbial analysis. Therefore, the objectives of this study were i) to investigate the properties of arugula, castor bean and chinaberry composts and vermicomposts and chitin on the root-knot nematode *M. javanica* under in vitro and glasshouse conditions, ii) to obtain information on the microbial structure of the best compost and vermicompost against the root-knot nematode, and iii) to determine the best combination of inhibitory plant vermicompost, chitin and beneficial bacteria in vermicompost.

## Materials and methods

### Ethics statement

The proposal for this research was approved by the Office of the Vice Chancellor for Academic Affairs and Graduate Studies at Shiraz University, and was conducted in the laboratories and glasshouse of the University's Plant Protection Department. No other permits were required to conduct this research. We also confirm that no endangered or protected species were involved in the studies.

## Preparing pure population of *Meloidogyne javanica*

Galled cucumber roots were collected from an infested greenhouse and carefully washed under running tap water to remove adhering soil particles. An egg mass was detached from the infected root using sterilized forceps, and the eggs were extracted and disinfected with 0.5% NaOCl [27]. Then, the eggs were placed in three holes around the roots of tomato seedling (*Lycopersicon esculentum* Mill. cv. Early Urbana) planted in a pot filled with pasteurized soil. The pot was kept under glasshouse conditions for 60 days [28]. After two months, the infected roots were removed from the soil and the eggs were extracted with sodium hypochlorite [27]. The pure culture was propagated in the same way on several tomato seedlings, and egg inoculum was obtained from infected tomato roots.

The pure population of the propagated nematode was identified by morphological characteristics including perineal patterns, and primers specific for *M. javanica*, *M. incognita*, and *M. arenaria* were used to confirm the morphological identification of the nematode species [29].

## Preparation of organic matter and bacteria

In this study, the required plants were provided by seed cultivation. Compost and vermicompost were prepared outdoors, and the bacteria were extracted from earthworms or vermicompost.

Seeds (local cultivars) of castor bean and arugula were planted in two plots. Then, the plants were harvested at flowering stage and used to produce compost and vermicompost. The green branches of chinaberry trees were collected in summer.

For the production compost, the shoots of castor bean and arugula, and the green branches of chinaberry were cut into pieces of 5 cm in diameter. Then, 10 kg of the chopped plants were placed in a 50-liter plastic barrel with 20 holes at the bottom to drain the leachate. The material was turned over every week with a garden fork to aerate it. They were stored for three months and watered as needed.

Vermicompost was made by adding the red worm *Eisenia foetida* to the chopped plant materials of castor bean, arugula and chinaberry. A plastic basket filled with vermicompost and worms, was placed in a barrel on top of the chopped plants. The worms gradually penetrated and fed on the plant tissues. After three months, the vermicompost was sieved and used for the experiments. Composted cow manure was also used to make the common vermicompost and as animal manure.

To prepare vermicompost extracts, the cotton bag containing the compost and vermicompost (100 g each) was immersed in a bucket of water (1 l) for two days at room temperature. Then, the water was strongly aerated with an air pump for one day [30].

The bacteria used in this study were *Bacillus safensis*, *B. megaterium*, *Pseudomonas resinovorans*, *Lysinibacillus* sp., *L. fusiformis* and *Sphingobacterium daejeonense*. They were isolated from earthworms or vermicompost, and their nematicidal activities were confirmed in previous study. The bacteria were cultured on nutrient agar (NA) and incubated for 48 hours at 28˚C in an incubator. After collecting the bacteria from the surface of the NA medium, they were dissolved in 1.5 ml of sterile distilled water by spinning and shaking. Then they were added to 50 ml of sterile distilled water until the desired bacterial concentration ($10^8$ CFU/ml) was determined with a spectrophotometer and 20 ml of it was used for the test steps [14].

Shrimp shells were purchased from the farmers' market. The collected shells were washed and dried. Then they were crushed with a grinding machine and used as a chitin source. Chitosan was extracted from the shrimp shells waste in three steps: demineralization, deproteinization and deacetylation [31].

## Evaluation of the effects of chitin and extracts from compost and vermicompost of inhibitory plants on *Meloidogyne javanica* under laboratory conditions

The effects of compost and vermicompost extracts of arugula, castor bean and chinaberry, animal manure and common vermicompost, and chitin on egg hatching and mortality of second-stage juveniles (J2s) of *M. javanica* were studied in vitro. About 100 eggs and 100 J2s in 1 ml of sterile water were placed in separate 6 cm Petri dishes. Then 5 ml of compost or vermicompost extracts were added. Distilled water was used as the control. Dead J2s were counted after 48 hours, and the unhatched eggs were counted after 72 hours [32]. The experiment was conducted as a completely randomized experimental design with three replicates.

## Evaluation of nematicidal effect of compost and vermicompost of inhibitory plants, bacteria and chitin on *Meloidogyne javanica* in the glasshouse

Four glasshouse experiments were conducted to evaluate the effects of different composts, vermicomposts, bacteria, and chitin on the root-knot nematode *M. javanica* in tomato roots. In the first experiment, the inhibitory effects of vermicompost and compost from castor bean, arugula and chinaberry, common vermicompost, animal manure, sulfur, shrimp shells and commercial chitosan on *M. javanica* activity were investigated. In the second experiment, the arugula vermicompost was used in combination with chitosan or the bacteria *Bacillus safensis*, *B. megaterium*, *Pseudomonas resinovorans*, *Lysinibacillus* sp., *L. fusiformis* and *Sphingobacterium daejeonense* as the most effective treatment. The bacteria were isolated from earthworms or vermicompost. In the third experiment, the effect of combining arugula vermicompost or chitosan with one or two bacterial isolates on root-knot nematode was investigated. This experiment was repeated in 10-kg pots.

**First experiment.**    This experiment was conducted in 3-kg plastic pots with a diameter of 19 cm, filled with pasteurized mixed soil (field soil and river sand in a ratio of 1:2). Based on the soil analysis, 40 mg P (kg soil)-1 as triple superphosphate and 60 mg N (kg soil)-1 as potassium nitrate were thoroughly mixed with the soil before sowing.

Sixty grams of different composts or vermicomposts (2% of potting soil) were homogeneously mixed with the soil in each pot. For chitosan treatment, 0.25 ml of commercial chitosan (Naturtrading Cia.Ltda, Ecuador) was added to each pot every 10 days. Sulfur (0.1 g) [33] and shrimp shells (0.5 g) were incorporated into the potting soil. Tomato seeds (cv. Early Urbana) were sown in the pots. Then, the roots of tomato seedlings at the four-leaf stage in half of the treated pots were inoculated with 6000 nematode eggs (two eggs per g of soil). Experiments were laid out on the glasshouse bench in a completely randomized design with four replicates. Pots were observed daily and watered as needed. The average maximum and minimum temperatures during the experiment were 32 and 26˚C, respectively. Plants were harvested 60 days after inoculation with the nematode, and their shoot fresh weight, dry weight, and root fresh weight were measured. Then, the J2s of the nematode were extracted from the soil of each pot using the tray method. For this purpose, a coarse-mesh plastic sieve was placed in a plastic tray as a base. A layer of paper towel was placed on top of it. A sample of 100 g of thoroughly mixed soil was evenly spread on the towel. Then enough water was added to the tray so that the sieve was slightly covered with water. The tray was kept at room temperature (25–28˚C) for 24 hours [34]. The J2s left the soil, passed through the paper towel and sank to the bottom of the dish. Then, the settled nematodes were collected using a 500-mesh sieve and counted. The number of galls, egg masses, and eggs in the roots of each plant were

also counted, and the final population (Pf) and the reproduction factor (Rf) of the nematode were calculated (Rf = Pf/Pi).

**Second experiment.** This experiment was similar to the first experiment, but the treatments included chitosan and the combination of arugula vermicompost with the bacteria *Bacillus safensis*, *B. megaterium*, *Lysinibacillus* sp., *L. fusiformis*, *Pseudomonas resinovorans* and *Sphingobacterium daejeonense* or chitosan. 60 g of arugula vermicompost was added to the potting soil and 0.05 g of chitosan was evenly mixed with the soil in the treatments. Then, tomato seedlings were inoculated at the four-leaf stage by adding 20 ml of the bacterial suspensions ($10^8$ CFU/ml) to the soil of each pot. Three days later, the roots of the seedlings were inoculated with 6000 eggs of *M. javanica* [14].

**Third experiment.** The conditions of this experiment were similar to those of the second experiment, but the combinations of arugula vermicompost and chitosan with effective bacteria were used as treatments.

**Fourth experiment.** This experiment was conducted similarly to the third, with five replicates and in 10 kg pots (25 cm diameter). For each pot, 200 g of vermicompost and 0.16 g of chitosan were used. Then, in half of the treated pots 20000 nematode eggs were inoculated around the roots of tomato plants at four-leaf stage. Eighty days after nematode inoculation, plant growth parameters and nematode indices were measured.

## Statistical analysis

The experimental data were analyzed using SAS statistical software (SAS 9.1). Parametric indices (plant indices) were analyzed using the Proc ANOVA method and non-parametric indices (nematode indices) were analyzed using the Friedman rank test method. Means were compared using a post-hoc Turkey's Honest Significant Difference (HSD) test (P < 0.05).

## Chemical analysis of compost and vermicompost of arugula

The chemical properties of the compost and vermicompost of arugula were determined by analyzing a sample of 500 g each. The pH, electrical conductivity (EC) and C/N ratio were measured. The pH and EC were measured using a pH/ EC meter in an aqueous suspension ratio of 1:10 (w/v). Total organic carbon (TOC) content was measured using a Shimadzu TOC-Vcp total organic carbon analyzer (Kyoto, Japan). Total nitrogen was determined by the Kjeldahl method [35].

## Bacterial community assessment, library construction and community sequencing of the arugula compost and vermicompost

The microbial DNA of the arugula compost and vermicompost, as the best treatment, was extracted from 0.5 g of each sample using the Top General Genomic DNA Purification Kit (Topaz Gene Research, Iran) according to the manufacturer's protocol. Five μl of the extracted DNA was placed on a 0.8% agarose gel and photographed using the Dock Gel device. The DNA concentration was measured using the NanoDrop 2000 spectrophotometer (Thermo Scientific, Inc., Waltham, MA, USA), and the DNA was then diluted to 5 ng/μl with sterile distilled water. The extracted and diluted DNA was stored at -20˚C in the freezer.

The V3-V4 region of the 16S rRNA gene was used to study the population structure of compost and vermicompost bacteria. The V3-V4 region of the 16S rDNA gene of the bacteria was amplified in two rounds using the primer pairs listed in S1 and S2 Tables. Amplification was performed using a thermal cycler, peqSTAR XS (Dresden, Germany). The PCR products from the first round were used for the second round. Five μl of the PCR products were added to a

1.5% agarose gel (containing 1.5 μl ethidium bromide per 100 ml gel) and loaded in an electric field at a voltage of 90 volts for 45 minutes in a TAE 1X buffer (S1 Fig).

DNA samples were prepared using the Illumina 16S Metagenomic Sequencing Library Preparation Protocol and the Nextera® XT DNA Index Kit6 (Illumina, USA). Samples were loaded onto a MiSeq reagent cartridge and then onto the instrument. A $2 \times 300$ bp paired-end sequencing run was performed. The resulting sequence reads were evenly distributed across the samples and showed uniform coverage. The Illumina MiSeq platform (Macrogen, Inc. South Korea, Seoul) was used in this study.

### Metagenomic analyses

CLC Genomics Workbench (version 20, QIAGEN, Venlo, The Netherlands) and the CLC Microbial Genomics Module plugin were used for metagenomic analysis. Quality control of reads was checked and all mismatches and fuzzy reads (N) were removed. Adaptors were trimmed from both ends of the reads. The total number of clean reads was 238776. Chimeras were removed, operational taxonomic units (OTUs) were clustered with a similarity level of 97%, and alpha and beta diversity were generated. A principal component analysis was also performed with a similarity of 3% level, which included the Chao1 index, Shannon index, Bray-Curtis dissimilarity and Jaccard index. Bacterial reads were classified using the reference database SILVA (v128) at an identity threshold of 97% [36].

## Results

### Effects of compost and vermicompost extracts from inhibitory plants on *Meloidogyne javanica* under laboratory conditions

Earthworms could not feed on the castor bean plant and its vermicompost was not produced. In this study, fresh shoots and leaves were used to produce compost and vermicompost. The fresh leaves of the plant cannot be used as earthworm feed because they tend to soften, rot, and release harmful exudates that are lethal to earthworms [37]. On the other hand, castor bean leaves have been shown to possess anthelmintic activity against the earthworm *Pheretima posthuma* [38]. For this reason, only its compost was used in this experiment. The results showed that the extracts of compost and vermicompost of arugula, common vermicompost, castor bean and chinaberry composts significantly affected the hatching of eggs and mortality of J2s of *M. javanica*. They inhibited nematode egg hatching by 12 to 32% and caused J2s mortality by 13 to 40%. The effects of animal manure, chinaberry vermicompost, and chitin (shrimp shells) on J2s and eggs were statistically similar to those of the control (Fig 1).

### The glasshouse experiments

**The nematicidal effect of compost and vermicompost of inhibitory plants, bacteria and chitin against *Meloidogyne javanica* in glasshouse (first experiment).** The result of the first experiment on the nematicidal effect of sulfur, sulfur + arugula compost, shrimp shells, commercial chitosan, animal manure, common vermicompost, common vermicompost + shrimp shells, and composts or vermicomposts of arugula, chinaberry and castor bean showed that there were no significant differences between the treatments and the control in the fresh weight of shoots of tomato plants. However, with the exception of sulfur, all other treatment significantly increased the dry weight of shoots of infected tomato plants (P ≤ 0.05). Moreover, vermicompost of arugula had the greatest effect on dry weight parameters (Table 1).

Among all treatments, vermicompost of arugula significantly reduced the final population and caused a 59.2% reduction in the reproduction factor of *M. javanica* (Table 1).

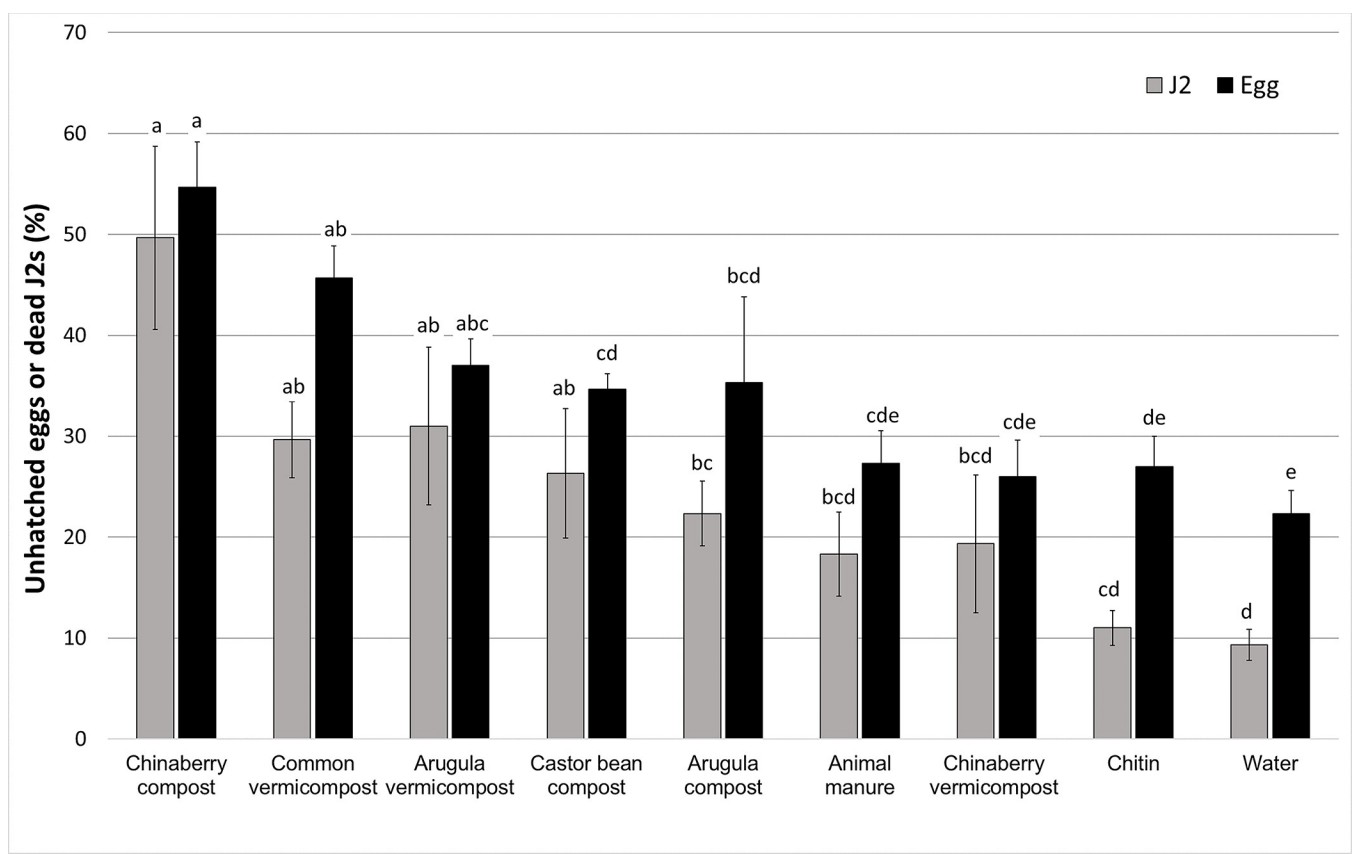

**Fig 1. Effects of different compost and vermicompost, and chitin (shrimp shells) on egg hatching and mortality of the second-stage juveniles (J2s) of** *Meloidogyne javanica* **in vitro.** Data are the means of three replicates. Bars with the same letters are not significantly different (P ≤ 0.05).

**The nematicidal effect of arugula vermicompost, bacteria and chitosan against** *Meloidogyne javanica* **in glasshouse (second experiment).** Similar to the first experiment, there were no significant differences (P ≤ 0.05) between treatments and control in the shoot fresh weight of tomato plants. The vermicompost of arugula with each of *Lysinibacillus* sp., *L. fusiformis*, *B. megaterium*, *S. daejeonense* or chitosan significantly increased the shoot dry weight of healthy tomato plants (without nematodes). Treatments of arugula vermicompost with *Lysinibacillus* sp., *B. megaterium*, or chitosan significantly increased the shoot dry weight of infected tomato plants (Table 2).

Data are means of four replicates. Means within a column with the same letter are not significantly different (P < 0.05).

The combination of arugula vermicompost with *B. megaterium* reduced the number of eggs in each egg mass. Chitosan and arugula vermicompost + *B. megaterium* reduced the number of nematode eggs in roots. Chitosan and the combination of arugula vermicompost with *P. resinovorans*, *B. megaterium* or chitosan significantly reduced the final population and caused a 67.9–70.9% reduction in the root-knot nematode reproduction factor (Table 2).

**The nematicidal effect of the combination of arugula vermicompost, bacteria and chitosan against** *Meloidogyne javanica* **in glasshouse (third experiment).** Shoot weight analysis of infected tomato plants showed that different bacteria together with arugula vermicompost and chitosan increased the shoot fresh weight of infected tomato plants, but there was no significant difference among treatments. In addition, the *P. resinovorans* + *S. daejeonense*

**Table 1. Effect of different compost and vermicompost, animal manure, shrimp shells and chitosan on the growth parameter of infected tomato plants and the nematode indices of *Meloidogyne javanica*.**

| Treatments | Plant growth parameters | | Root fresh weight[b] (g) | Nematode indices | | |
| --- | --- | --- | --- | --- | --- | --- |
| | Shoot dry weight (g) | | | | | |
| | With nematode | Without nematode | With nematode | Final population[c] | Reproduction factor (Rf) | Rf reduction (%)[d] |
| Sulfur | 3.05 k | 5.1 j | 5.62 | 17304 bc | 2.88 bc | 49.2 |
| Sulfur+ arugula compost | 8.57 f-h | 9 c-h | 11.42 | 19287 abc | 3.21 abc | 43.3 |
| Arugula compost | 8.7 e-h | 8.2 f-h | 16.7 | 17328 abc | 2.88 abc | 49.1 |
| Arugula V.[a] | 10.87 a-d | 11.5 ab | 22.6 | 13875 c | 2.31 c | 59.2 |
| Chinaberry compost | 8.82 e-h | 10.9 a-d | 14.3 | 24802 abc | 4.13 abc | 27.1 |
| Chinaberry V. | 9.5 b-g | 11.6 a | 18.6 | 31331 ab | 5.22 ab | 8.0 |
| Castor bean compost | 9.75 a-f | 11 a-c | 22.02 | 33156 a | 5.52 a | 2.6 |
| Common V. + shrimp shells | 7.42 hi | 9.6 a-f | 14.5 | 16777 bc | 2.79 bc | 50.7 |
| Common V. | 9.1 c-h | 10 a-f | 14.15 | 17252 abc | 2.87 abc | 49.3 |
| Animal manure | 9.62 a-f | 10.6 a-e | 18.85 | 34398 a | 5.73 a | - |
| Shrimp shells | 7.2 hi | 8.95 d-h | 14.45 | 20841 abc | 3.47 abc | 38.8 |
| Commercial chitosan | 6.2 ij | 7.52 g-f | 9.35 | 19506 abc | 3.2 abc | 42.7 |
| Distilled water (control) | 2.52 k | 5.17 j | 7.57 | 34042 ab | 5.67 ab | - |

[a]: V. = vermicompost; [b]: The treatments had no significant effect on the root fresh weight of infected plants.; [c]: Eggs + J2s + females; [c]: Reduction percent of nematode reproduction factor compared to the control.

Data are means of four replicates. Means within a column with the same letter are not significantly different (P < 0.05).

**Table 2. Effect of arugula vermicompost + bacteria on the growth parameter of infected tomato plants and the nematode indices of *Meloidogyne javanica*.**

| Treatments | Plant growth parameters | | Root fresh weight[b] (g) | Nematode indices | | | | |
| --- | --- | --- | --- | --- | --- | --- | --- | --- |
| | Shoot dry weight (g) | | | | | | | |
| | With nematode | Without nematode | With nematode | Eggs/ root | Eggs/egg mass | Final population[c] | Reproduction factor (Rf) | Rf reduction (%)[d] |
| Arugula V.[a] + *Bacillus safensis* | 7.33 b-e | 7.5 b-e | 17.7 | 80113 abc | 42 ab | 97562 ab | 16.26 ab | 27.8 |
| Arugula V. + *Pseudomonas resinovorans* | 8.16 def | 8.53 a-e | 20.34 | 34800 bc | 28 ab | 41185 bc | 6.86 bc | 69.5 |
| Arugula V. + *Lysinibacillus* sp. | 10.43 a-d | 13.76 a | 18.33 | 60100 abc | 248 ab | 70943 abc | 11.82 abc | 47.5 |
| Arugula V. + *Lysinibacillus fusiformis* | 8.73 a-e | 9.2 a-d | 15.33 | 97408 ab | 75 ab | 111845 a | 18.64 a | 17.2 |
| Arugula V. + *Bacillus megaterium* | 9.53 a-d | 12.23 ab | 20.76 | 31718 c | 19 b | 39291 c | 6.54 c | 70.9 |
| Arugula V. + *Sphingobacterium daejeonense* | 6.16 de | 10.46 a-d | 13.33 | 139307 a | 146 ab | 146121 a | 24.35 a | - |
| Arugula V. + chitosan | 10 a-d | 12.9 ab | 23.56 | 34481 bc | 43 ab | 39540 bc | 6.59 bc | 70.7 |
| Chitosan | 6.7 c-e | 7.2 b-e | 6.9 | 25815 c | 136 a | 43335 bc | 7.22 bc | 67.9 |
| Distilled water (control) | 2.83 e | 6.3 c-e | 4.1 | 97807 ab | 187 a | 135049 a | 22.5 a | - |

[a]: V. = vermicompost

[b]: The treatments had no significant effect on the root fresh weight of infected plants.

[c]: Eggs + J2s + females; [c]: Reduction percent of nematode reproduction factor compared to the control.

**Table 3. Effect of arugula vermicompost + combination of bacteria and chitosan on the nematode indices of *Meloidogyne javanica* and the growth parameter of infected tomato plants.**

| Plant growth parameters | | | | | |
|---|---|---|---|---|---|
| | Shoot fresh weight (g) | | Shoot dry weight (g) | | Root fresh weight (g) |
| Treatments | With nematode | Without nematode | With nematode | Without nematode | With nematode |
| Arugula V.[a] + *Pseudomonas resinovorans* + chitosan | 48 a | 54.25 a | 8.75 a | 9.5 a | 18 |
| Arugula V. + *P. resinovorans* + *Sphingobacterium daejeonense* + chitosan | 51.25 a | 57.75 a | 10 a | 10.75 a | 17.75 |
| Arugula V. + *Bacillus megaterium* + chitosan | 45.75 a | 59 a | 8 abc | 10.25 a | 18.5 |
| Arugula V. + *B. megaterium* + *Lysinibacillus* sp. + chitosan | 49.75 a | 57.5 a | 8.25 ab | 10.2 a | 20.75 |
| Distilled water (control) | 19 b | 26.75 b | 4.2 bc | 4 b | 15.75 |
| **Nematodes indices** | | | | | |
| Treatments | Eggs/root | Eggs/Egg mass | Final population[c] | Reproduction factor (Rf) | Rf reduction (%)[d] |
| Arugula V. + *Pseudomonas resinovorans* + chitosan | 166180 ab | 189 a | 168602 ab | 28.1 ab | 54.0 |
| Arugula V. + *P. resinovorans* + *Sphingobacterium daejeonense* + chitosan | 82640 b | 64 b | 85901 b | 14.31 b | 76.6 |
| Arugula V. + *Bacillus megaterium* + chitosan | 140530 b | 67 ab | 143423 b | 23.9 b | 60.9 |
| Arugula V. + *B. megaterium* + *Lysinibacillus* sp.+ chitosan | 166750 ab | 66 b | 170391 ab | 28.39 ab | 53.6 |
| Distilled water (control) | 359050 a | 146 a | 366802 a | 61.13 a | - |

[a]: V. = vermicompost

[b]: The treatments had no significant effect on the root fresh weight of infected plants.

[c]: Eggs + J2s + females; [c]: Reduction percent of nematode reproduction factor compared to the control.

Data are means of four replicates. Means within a column with the same letter are not significantly different (P < 0.05).

+ arugula vermicompost + chitosan and *P. resinovorans* + arugula vermicompost + chitosan treatments were significantly different (P ≤ 0.05) from the control in terms of shoot dry weight of infected plants. All treatments significantly (P ≤ 0.05) increased the fresh weight and dry weight of the shoots of uninfected plants (Table 3).

The combinations of arugula vermicompost + *P. resinovorans* + *S. daejeonense* + chitosan and arugula vermicompost + *B. megaterium* + *Lysinibacillus* sp. + chitosan significantly decreased the number of eggs in each egg mass. Among all combinations of bacteria + arugula vermicompost + chitosan, both arugula vermicompost + *P. resinovorans* + *S. daejeonense* + chitosan and *B. megaterium* + arugula vermicompost + chitosan treatments significantly reduced eggs in roots, the final population and reproduction factor of the root-knot nematode. Arugula vermicompost + *P. resinovorans* + *S. daejeonense* + chitosan had the greatest effect, reduced the nematode reproduction factor by 76.6% (Table 3).

**The nematicidal effect of the combination of arugula vermicompost, bacteria and chitosan against Meloidogyne javanica in the glasshouse (fourth experiment).** Arugula vermicompost and all combinations of bacteria + arugula vermicompost + chitosan significantly (P ≤ 0.05) increased shoot fresh weight and fruit weight of infected tomato plants. Except for arugula vermicompost + *B. megaterium* + chitosan and arugula vermicompost, the other treatments including, arugula vermicompost + *P. resinovorans* + chitosan, arugula vermicompost + *P. resinovorans* + *S. daejeonense* + chitosan and arugula vermicompost + *B. megaterium* + *Lysinibacillus* sp. + chitosan, also significantly (P ≤ 0.05) increased shoot dry weight. None of the treatments negatively affected the growth indices of the uninfected plants and resulted in a relative or significant increase in these indices (Table 4).

**Table 4. Effect of arugula vermicompost + combination of bacteria and chitosan on the nematode indices of *Meloidogyne javanica*, and growth parameter and yield of infected tomato plants in 10 kg pots.**

| Plant growth parameters | | | | | | |
|---|---|---|---|---|---|---|
| **Treatments** | Shoot fresh weight (g) | | Shoot dry weight (g) | | Fruit weight (g) | |
| | With nematode | Without nematode | With nematode | Without nematode | With nematode | Without nematode |
| Arugula V.[a] + *Pseudomonas resinovorans* + chitosan | 224 ab | 232 ab | 49.7 ab | 49.9 ab | 153 bc | 261 a |
| Arugula V. + *P. resinovorans* + *Sphingobacterium daejeonense* + chitosan | 255 a | 253 a | 53.9 a | 47.5 ab | 200 abc | 276 a |
| Arugula V. + *Bacillus megaterium* + chitosan | 179 b | 236 ab | 36.3 bc | 50.5 a | 170 bc | 160 bc |
| Arugula V. + *B. megaterium* + *Lysinibacillus* sp.+ chitosan | 219 ab | 216 ab | 50.9 a | 50.9 a | 150 bc | 237 ab |
| Arugula V. | 214 ab | 226 ab | 41.8 abc | 50.3 a | 197 abc | 230 ab |
| Distilled water (control) | 112 c | 180 b | 28.8 c | 40.4 abc | 42.6 d | 145 c |

| Nematodes indices | | | | | |
|---|---|---|---|---|---|
| **Treatments** | Egg mass/root | Eggs/root | Final population[c] | Reproduction factor (Rf) | Rf reduction (%)[d] |
| Arugula V. + *Pseudomonas resinovorans* + chitosan | 4500 b | 377316 b | 402816 b | 20.1 b | 58.9 |
| Arugula V. + *P. resinovorans* + *Sphingobacterium daejeonense* + chitosan | 4532 b | 332577 b | 361531 b | 18.1 b | 63.1 |
| Arugula V. + *Bacillus megaterium* + chitosan | 6819 ab | 503187 ab | 601599 ab | 30.1 ab | 38.7 |
| Arugula V. + *B. megaterium* + *Lysinibacillus* sp. + chitosan | 11267 a | 551569 ab | 615374 ab | 30.8 ab | 37.3 |
| Arugula V. | 7544 ab | 509161 ab | 560158 b | 28 b | 43.9 |
| Distilled water (control) | 11504 a | 851777 a | 980885 a | 49 a | - |

[a]: V. = vermicompost

[b]: The treatments had no significant effect on the root fresh weight of infected plants.

[c]: Eggs + J2s + females; [c]: Reduction percent of nematode reproduction factor compared to the control.

Data are means of five replicates. Means within a column with the same letter are not significantly different (P < 0.05).

The arugula vermicompost + *P. resinovorans* + chitosan and arugula vermicompost + *P. resinovorans* + *S. daejeonense* + chitosan treatments significantly ($P \leq 0.05$) decreased the number of egg masses and eggs in the roots. These treatments and the vermicompost of arugula significantly reduced the final population and caused a 43.9–63.1% reduction in the reproduction factor of *M. javanica* (Table 4).

## Chemical properties of compost and vermicompost of *E. sativa*

Chemical analysis showed that the compost of arugula had higher pH, electrical conductivity (EC) and C/N ratio than the vermicompost (compost: pH = 8.4, EC = 3.3 ds/m; C/N ratio = 31.83, vermicompost: pH = 7.3, EC = 0.73 ds/m; C/N ratio = 22.9).

## Bacterial diversity analysis of arugula compost and vermicompost

The 16S rRNA gene of bacteria isolated from compost and vermicompost of arugula was analyzed. Two samples of the arugula vermicompost and compost were clustered using the Bray-Curtis dissimilarity metric and Jaccard index (PERMANOVA p = 0.0001). Fig 2A and 2B shows two groups of vermicompost and compost, each forming a separate group. According to component 1 (PCo1; the total variance of OTUs is 81% and 71%), the microbiota of compost is significantly different from vermicompost. Differential abundance analysis (S2 Fig) also showed that the bacterial density of vermicompost samples was different from that of compost. The vermicompost samples were similar in their microbial communities, and the compost samples were too.

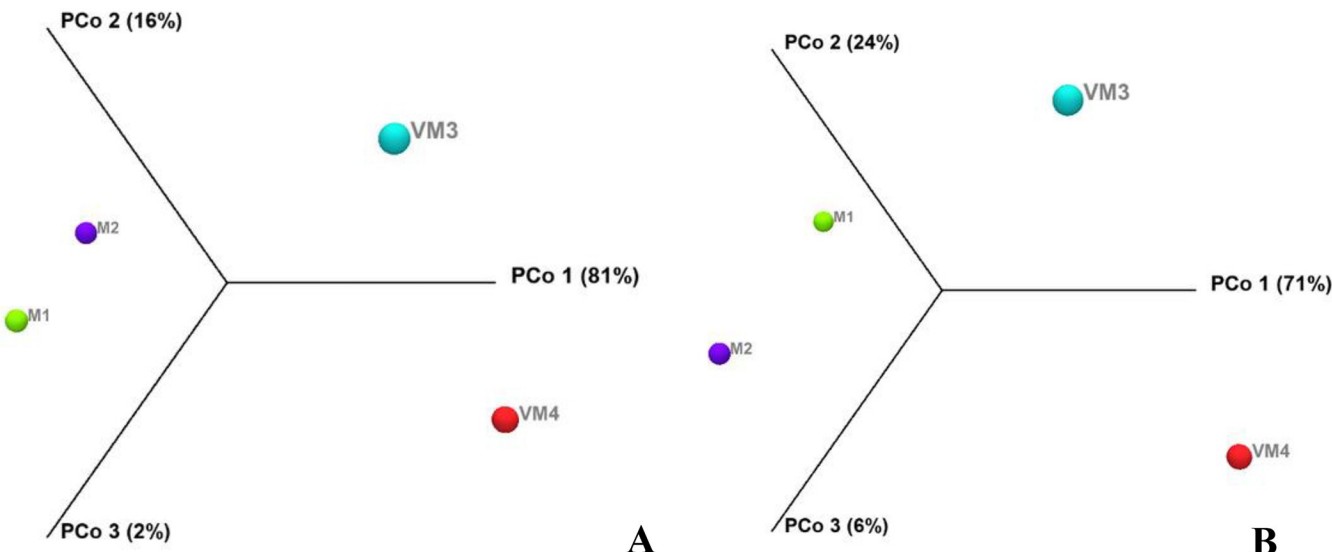

**Fig 2.** Principal component analysis (PCA) of compost and vermicompost of arugula; beta diversity estimated by Bray-Curtis dissimilarity (A) and Jaccard index (B) (M: Arugula compost; VM: Arugula vermicompost).

Alpha diversity was estimated for all vermicompost and compost samples. Chao1 and Shannon diversity indices were higher in the vermicompost samples than in the compost samples (Fig 3), indicating that the bacterial community of vermicompost was richer than that of compost. The processing of vermicompost also influenced bacterial diversity.

The bacterial community of vermicompost and compost differed at the phyla level (Fig 4). The most abundant phylum in the compost was *Firmicutes*, whereas in the vermicompost it was *Proteobacteria*. In addition, the number of members of *Actinobacteria*, *Chloroflexi*, *Acidobacteria*, and *Gemmatimonadetes* is higher in the vermicompost than in the compost.

*Chitinophagaceae*, *Hyphomicrobiaceae* and *Lachnospiraceae* in vermicompost, and *Bacillaceae*, *Clostridiaceae*, *Paenibaccillaceae* and *Planococcaceae* in compost were the most bacterial taxa at the family level, indicating the difference between vermicompost and compost (Fig 5).

*Proteobacteria* is the most important phylum in the bacterial community of vermicompost, accounting for 20–42% (257–293 OTUs) in vermicompost and 24–34% (190–196 OTUs) in compost. After *Proteobacteria*, *Firmicutes* was the next dominant phylum with 6–22% in vermicompost and 50–62% in compost. The predominant families or orders (suborders) of the phyla *Proteobacteria* and *Firmicutes* in arugula compost and vermicompost are shown in Table 5.

Other five taxa, that showed the difference between vermicompost and compost were *Actinobacteria* (22% and 71–112 OTUs in vermicompost, 2–3% and 30–44 OTUs in compost), *Bacteroidetes* (12–18% and 78–93 OTUs in vermicompost, 5–6% and 73–79 OTUs in compost) (Table 5), *Gemmatimonadetes* (4% and 16–26 OTUs in vermicompost, <1% and 4–8 OTUs in compost), *Chloroflexi* (3–4% and 40–81 OTUs in vermicompost, <1% and 13–15 OTUs in compost), *Acidobacteria* (3–4% and 29–51 OTUs in vermicompost, <1% and 11 OTUs in compost).

The families of *Bacteroidetes* of vermicompost and compost were *Chitinophagaceae*, *Chryseolinea*, *Sphingobacteriaceae*, *Rhodothermaceae*, *Cytophagaceae*, *Flavobacteriaceae*, *Prolixibacteraceae*, *Cryomorphaceae*, *Saprospiraceae*, and *Marinilabiliaceae*.

All genera of the four samples are shown in Fig 6. Nine of them were the most abundant. The bacterial genera in the compost were *Bacillus*, *Clostridium*, *Paenibacillus* and

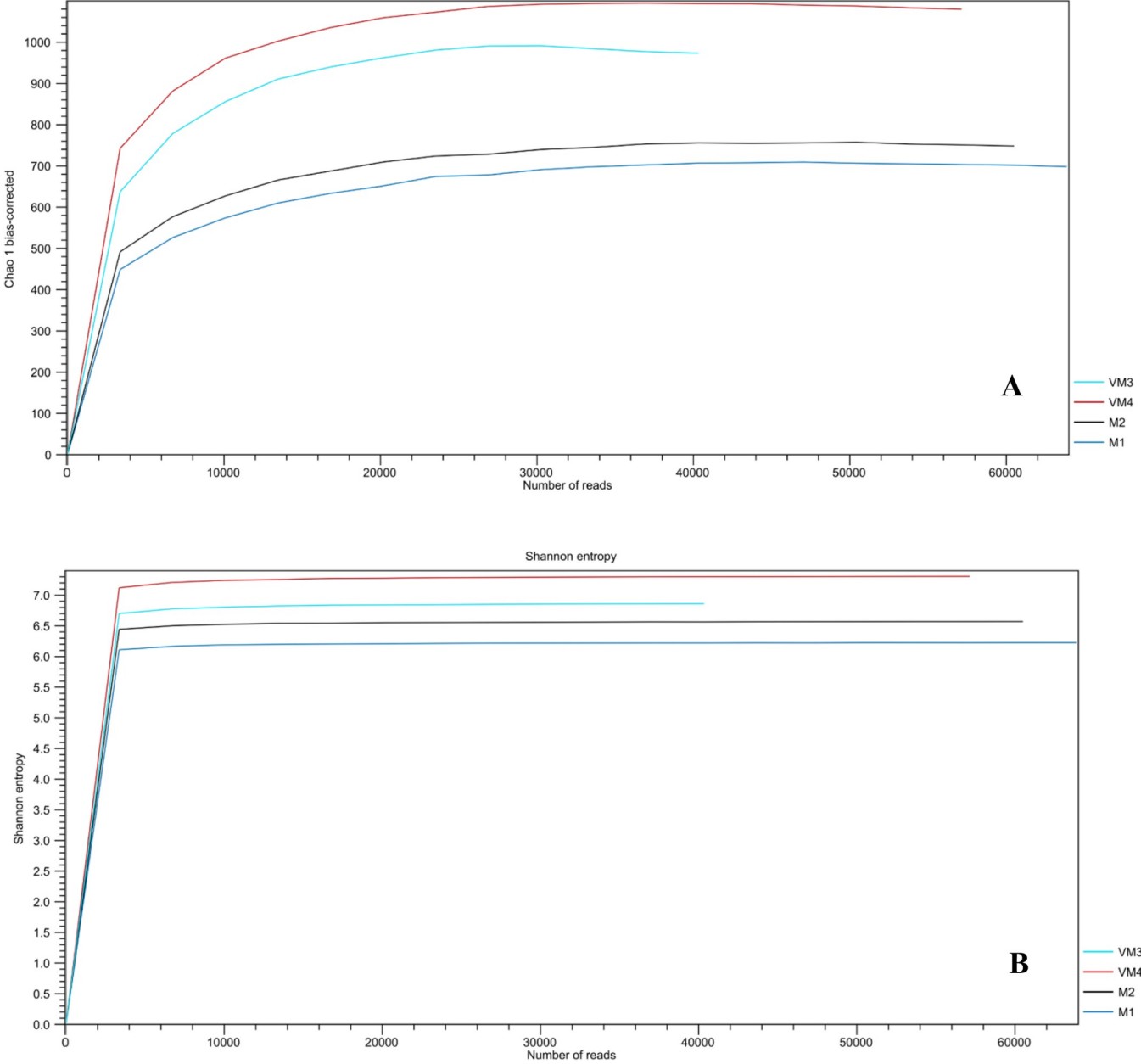

**Fig 3.** Alpha diversity estimates (A: Chao1 index; B: Shannon index) for the bacterial community of vermicompost and compost of arugula (M: Arugula compost; VM: Arugula vermicompost).

*Psudoxanthomonas*, and in the vermicompost were *Geodermatophilus*, *Thermomonas*, *Lewinella*, *Pseudolabrys* and *Erythrobacter*.

## Discussion

The results of the present study in the first glasshouse experiment showed that all organic materials used in this study increased the dry weight of shoots of infected and healthy tomato plants, except for the sulfur treatment. Similar to our previous studies, the plant dry weight

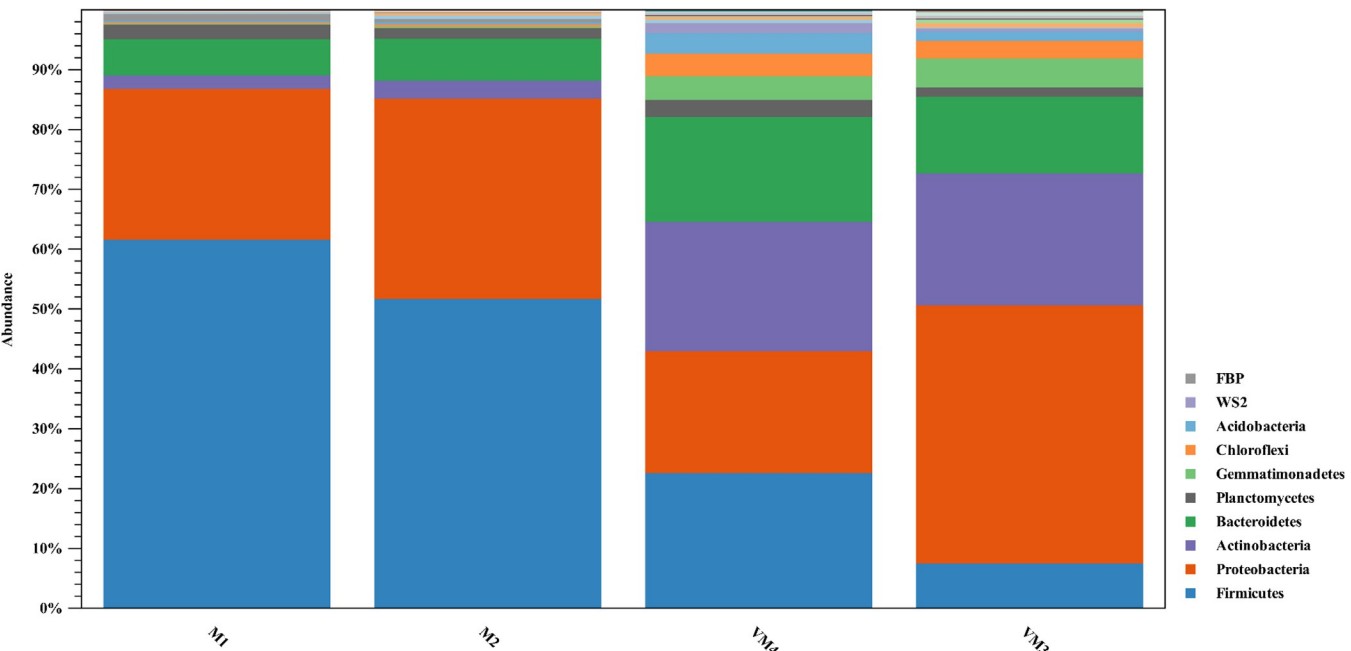

**Fig 4. Taxonomic profile of bacteria in compost and vermicompost of arugula at the phylum level (M: Arugula compost; VM: Arugula vermicompost).**

index compared to the fresh weight index better defined the difference between treatments and their effects [30,39]. Only the arugula vermicompost caused a significant reduction in the reproduction factor of the root-knot nematode *Meloidogyne javanica*. Arugula compost alone or with sulfur, chinaberry compost and common vermicompost, and shrimp shells alone or in combination also decreased the reproduction factor, but there was no significant difference

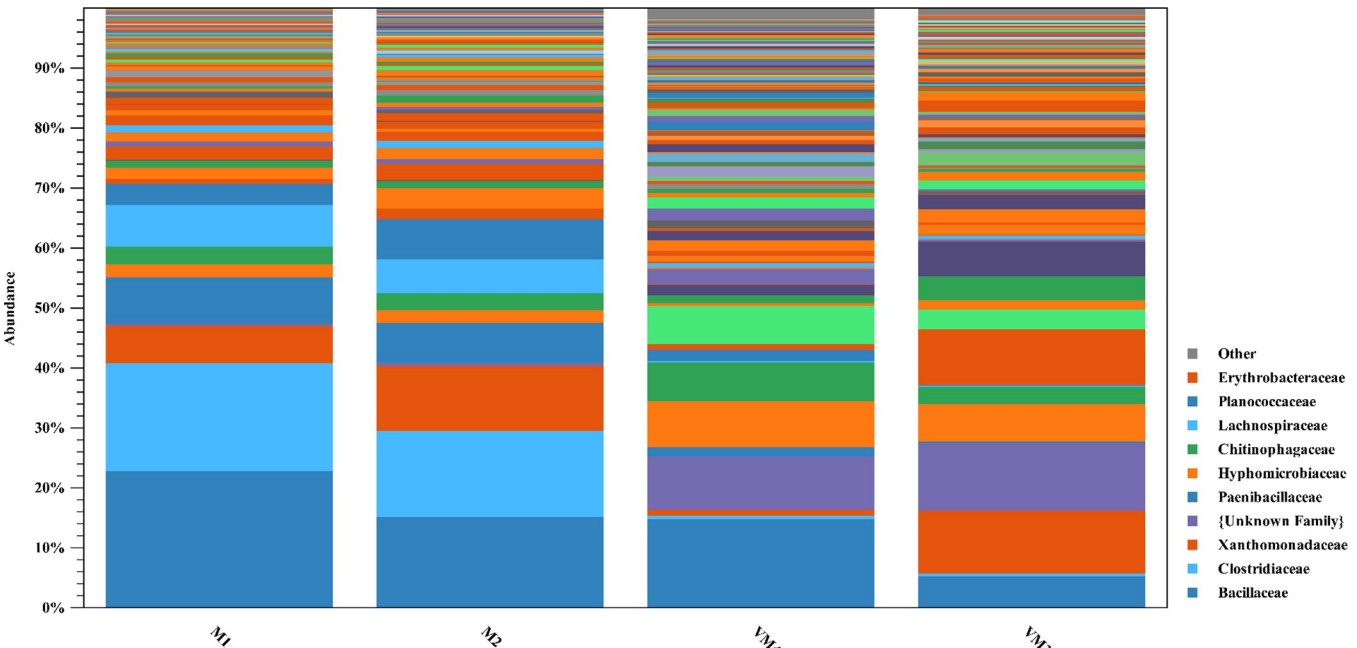

**Fig 5. Taxonomic profile of bacteria in compost and vermicompost of arugula at the family level (M: Arugula compost; VM: Arugula vermicompost).**

**Table 5. The predominant bacterial families or orders (suborders) were isolated from arugula compost and vermicompost.**

| | Phyla | | |
|---|---|---|---|
| Compost | *Proteobacteria* | *Firmicutes* | *Actinobacteria* |
| vermicompost | 20–42%<br>*Rhizobiales*<br>*Sorangiineae*<br>*Cystobacterineae*<br>*Hyphomicrobiaceae*<br>*Rhodospirillaceae*<br>*Syntrophobacteraceae*<br>*Xanthomonadaceae* | 6–22%<br>*Bacillaceae*<br>*Clostridiaceae*<br>*Heliobacteriaceae*<br>*Lachnospiraceae*<br>*Paenibacillaceae*<br>*Ruminococcaceae*<br>*Syntrophomonadaceae*<br>*Thermoactinomycetaceae*<br>*Thermoanaerobacteraceae*<br>*Veillonellaceae* | 22%<br>*Acidimicrobiales*<br>*Actinomycetales*<br>*corbibacteriales*<br>*Euzebyales*<br>*Gaiellales*<br>*Nitriliruptorales*<br>*Solirubrobacterales*<br>*Thermoleophilaceae* |
| compost | 24–34%<br>*Caulobacteraceae*<br>*Comamonadaceae*<br>*Coxiellaceae*<br>*Hyphomicrobiaceae*<br>*Hyphomonadaceae*<br>*Sinobacteraceae*<br>*Sphingomonadaceae*<br>*Xanthomonadaceae* | 50–62%<br>*Bacillaceae*<br>*Clostridiaceae*<br>*Gracilibacteraceae*<br>*Heliobacteriaceae*<br>*Lachnospiraceae*<br>*Paenibacillaceae*<br>*Ruminococcaceae* | 2–3%<br>*Acidimicrobiales*<br>*Actinomycetales*<br>*Solirubrobacterales* |

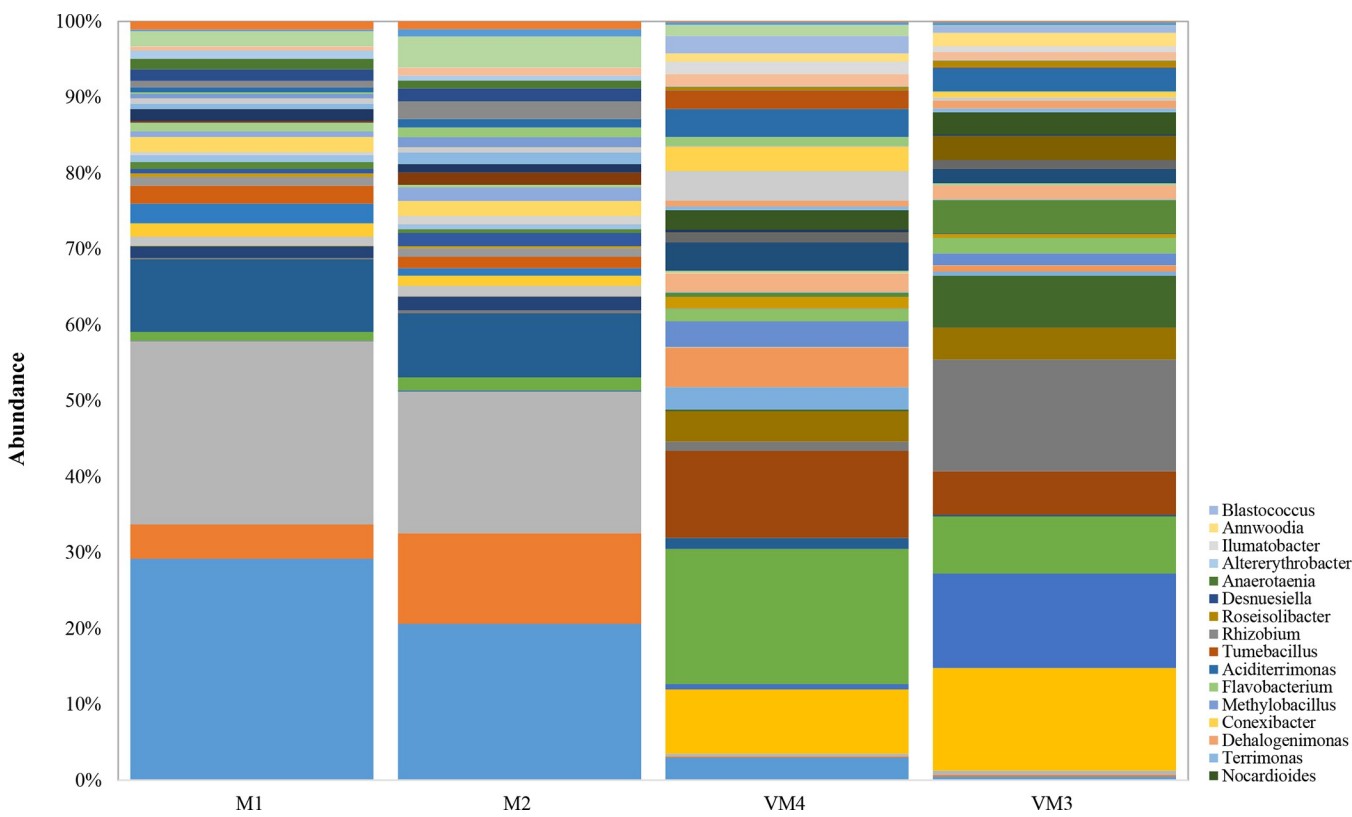

**Fig 6. Taxonomic profile of bacteria in compost and vermicompost of arugula at the genus level (M: Arugula compost; VM: Arugula vermicompost).**

from the control. Previous studies have shown that all parts of chinaberry and castor bean seeds have nematicidal properties [40,41], but in this study their compost or vermicompost was not effective in reducing nematode damage. Animal manure also failed to reduce the final nematode population in the first experiment. Oka [42] reported that of the animal manures, chicken litter resulted in an increase in soil microbes and a decrease in the *M. incognita* population, which is one of the most effective manures for controlling root-knot nematode damage. Also, the use of sulfur and sulfur + arugula compost did not reduce the nematode reproduction factor and did not increase the dry weight of the plants. Saffari *et al*. [40] reported that vermicompost stimulated the growth of *Scindapsus aureus* more than granular sulfur compost, prepared by adding sulfur to sugar beet molasses and compost garbage. This result could be due to the fact that vermicompost absorbs more nutrients and has better nutrient properties.

Treatments with common vermicompost + shrimp shells (representative of chitin) and commercial chitosan decreased the number of J2s in potting soil. Our result is in agreement with the report of Castro *et al*. [22], who showed that the addition of vermicompost + chitin to potting soil reduced the reproduction factor of *M. incognita*. Spiegel *et al*. [41] also reported that the gall index of *M. javanica* decreased after the application of chitin, although this treatment did not change the fresh weight of the shoots of infected plants compared to the control. Chitin and its derivatives can increase the population of chitinolytic microorganisms, such as *Telluria chitinolytica*. This bacterium was isolated from the shells of crustaceans and significantly affected the J2s of *M. javanica* [43]. Also, chitin stimulates the plant defense system and improves plant growth, and its decomposition adds various toxic compounds such as ammonia to the soil [44]. Due to ion absorption and GSH binding of chitin and its derivatives, they have nematicidal properties [45]. In addition, chitosan is an important chitin derivative that has been shown to reduce nematode penetration in tomato roots in this study. Chitosan contains nitrogen, which is toxic to nematodes and strengthens plant resistance to disease [46], and is used as a colonization enhancer of *Pochonia chlamydosporia* in soil against root-knot nematode [47]. In the present study, chitosan extracted from shrimp shells was combined with vermicompost to enhance the effect of vermicompost and chitosan.

The arugula vermicompost used in this study was the best compost for reducing the nematode population while increasing plant growth. Therefore, it was selected from the first glasshouse experiment for the other experiments. This plant belongs to the *Brassicaceae* family, whose cultivation suppresses soil-borne diseases such as nematodes [48]. This property is caused by glucosinate and non-glucosinate (sulfur-containing) compounds, and their use in soil alters biological and physiological factors. Nematodes suppression has been demonstrated by using members of this plant family such as rapeseed (*Brassica napus*) and Indian mustard (*Brassica juncea*) [42]. The nematicidal effect of *Brassicaceae* is related to the production of organic cyanides, nitriles and the increase of nematode antagonists. Arugula, a member of the *Brassicaceae* family, has nematicidal effect as a soil amendment and stimulates plant root growth [49]. There is no report of control of nematode damage by vermicompost of arugula.

The addition of bacteria isolated from earthworms and vermicompost increased the inhibitory activity of arugula vermicompost against the root-knot nematode. Some species of the genera *Agrobacterium*, *Alcaligenes*, *Arthrobacter*, *Bacillus*, *Enterobacter*, *Erwinia*, *Pseudomonas*, *Rhizobium*, *Serratia*, *Stenotrophomonas*, *Streptomyces* and *Xanthomonas* are known as biological control agents of plant pathogens. Cost-effective mass production on an industrial scale has been developed for some of these bacteria [50]. Since the bacteria used in this study belong to the genera used for the production of commercial formulations and their inhibitory properties were demonstrated in our previous study [14], they have the potential for commercial production. Although all six species *Bacillus safensis*, *B. megaterium*, *Lysinibacillus* sp., *L. fusiformis*, *Pseudomonas resinovorans* and *Sphingobacterium daejeonense* successfully reduced

nematode populations when combined with vermicompost, only *P. resinovorans* and *B. megaterium* showed satisfactory effects in the present study. The difference in the effectiveness of the bacteria probably depends on their survivability and the level of inoculum. Bacterial populations in vermicompost are determined by their competitiveness [51] and depend on the microbial community of the vermicompost. The result of enrichment of vermicompost with the optimized level of the bacteria *Azospirillum brasilense* and *Rhizobium leguminosarum* showed a significant negative correlation between the population and the storage period, and the total bacterial populations were high at the beginning of incubation and then decreased towards the end [52]. Moreover, other predators and bacteriophages can alter the microbial communities that are live antibiotics in the gut of the worms and affect the bacteria [53]. Although *Lysinibacillus* sp. and *S. daejeonense* failed to reduce nematode populations, they improved plant growth indices. Therefore, a combination of *P. resinovorans* + *S. daejeonense* (gram-negative) and *B. megaterium* + *Lysinibacillus* sp. (gram-positive) with arugula vermicompost and chitosan was used as the most effective treatment. The combination of gram-negative bacteria increased not only plant growth indices but also fruit weight. Importantly, arugula vermicompost + *P. resinovorans* + *S. daejeonense* + chitosan was the best combination to reduce the nematode population (up to 63.1%). Abdel-Salam *et al*. [54] showed that protoplast fusion of *Bacillus amyloliquefaciens* and *Lysinibacillus sphaericus* increased yield production, chitin enzyme activity, and nematicidal potential of bacteria. In addition, the mixture of *Mycobacterium* and *Rhizobium* had a synergistic effect on juvenile mortality of *M. incognita* [55]. In general, the biological control potential of combination agents was enhanced by expanding the spectrum of activity for other pathogens, duplicating modes of action, targeting more than one stage of the pathogen life cycle, and stabilizing each other in soil [56]. Thus, our results suggest that the vermicompost of arugula is an excellent base for combining with bacteria and chitosan that retain their properties.

The present result confirms that vermicompost of arugula has better chemical properties than its compost. Earthworms improve the properties of organic material to produce compost, and vermicompost is superior to compost in producing plant biomass [57]. Thus, vermicomposting of arugula increases the content of N, P, Fe and Mg, and lowers pH [39]. Due to the nematicidal nature of arugula plants, vermicompost production increased the utility and quality of vermicompost and arugula plants as soil amendments. Moreover, earthworms have been reported to increase aeration, substrate fragmentation and microbial activities during the vermicomposting process [58], and vermicompost is more homogeneous than compost and has better chemical properties [59]. Also, proper compost particle size increases microbial biomass [60]. Therefore, understanding the microbial structure of vermicompost may clarify its ability to replace chemical fertilizers and nematicides.

The present study showed that the microbial richness of arugula vermicompost was higher than that of arugula compost based on alpha diversity analysis. Beta diversity and heat map also showed that the bacterial community of vermicompost was different from that of compost samples.

*Proteobacteria* are the most abundant phylum in vermicompost [18]. *Myxococcales* as an order of this phylum accounted for 7–15% of *Proteobacteria* in the vermicompost of arugula in the present study. This bacterial order includes genera that produce antifungal and antibiotic metabolites. The polyketide soraphen A and chivosazol are antifungal agents produced by *Sorangium cellulosum*, a genus of *Myxococcales* [61,62]. *Rhodospirillaceae* were 9% of the *Proteobacteria* in the vermicompost of arugula. They are non-sulfur purple photosynthetic bacteria and a group of nitrogen fixers [63]. The genus *Azospirillum* from this bacterial family produces phytohormones and has plant growth-promoting properties [64]. *Hyphomicrobiaceae* accounted for 6–7% of the *Proteobacteria* in the vermicompost of arugula in the present study.

They can proliferate in the root zone of plants and participate in soil nitrogen cycling [65]. *Devosia neptuniae*, the species of this family, is can form nitrogen-fixing a symbiosis with legumes [66]. In addition, *Bdellovibrio* is one of the most important genera of *Deltaproteobacteria* found only in the vermicompost of arugula. This genus is a gram-negative bacterial predator such as *Burkholderia cepacia*, *Pectobacterium atrosepticum*, and *Pseudomonas glycinea* [67–69]. In this study, *Thermomonas*, *Pseudolabrys* and *Erythrobacter* were the most abundant genera of *Proteobacteria*. *Thermomonas* is an N-fixing bacterium, and *Pseudolabrys* belongs to the *Rhizobiales* and interacts with plant roots [70]. In addition, *Erythrobacter* has an antagonistic ability against the causal agent of tomato wilt [71].

*Actinobacteria* is a phylum highly abundant in vermicompost of arugula after *Proteobacteria*, which is consistent with the report of Cai *et al*. [18]. They reported that *Actinobacteria* was the most abundant phylum in the vermicompost after *Proteobacteria*. These bacteria decompose organic matter and complex mixtures such as cellulose and chitin. One genus of this phylum is *Geodermatophilus*, which has this ability [72] and was abundant in the vermicompost samples of arugula. In addition, earthworms promoted the abundance of *Bacteroidetes* and *Gemmatimonadetes* in the vermicomposting process [73], and both were more abundant in the vermicompost of arugula than in the compost. In previous studies, *Gemmatimonadetes* was detected on J2s of the root-knot nematode in the suppressing soil [74]. The genera of *Bacteroidetes* degrade macromolecules, including cellulose and chitin [75]. The genus *Lewinella* can degrade complex organic material [76]. It was abundant in the vermicompost of arugula. *Chitinophagaceae* is an important chitinolytic bacterial family that was more abundant in arugula vermicompost than in compost. Members of the *Chitinophagaceae* are antagonists of plant pathogens. They have the potential for antifungal and nematicidal activity since chitin is the main component of nematode egg shell and fungal structures [77]. Moreover, *Actinobacteria* is characterized by the production of Indole-3-acetic acid (IAA) and antibiotics, nitrogen fixation ability, and biocontrol control of plant pathogens [78]. In addition to these properties, this phylum has the potential to inhibit hatching of root-knot nematode eggs [79]. *Streptomyces* is a genus that produces avermectin, an antibiotic against nematodes [80]. In addition, a previous report found that earthworm casts fed cow manure had a higher abundance of *Chloroflexi* than pig and horse manures [81]. In this study, *Anaerolineaceae* (28–40% of *Chloroflexi* in vermicompost of arugula) and *Caldilineaceae* (30–45% of *Chloroflexi* in vermicompost of arugula) were the dominant families. *Anaerolineaceae* degrades organic matter under anoxic conditions and is involved in Cd solubility [82]. Soil application of raw garlic stalks increased beneficial bacteria such as *Anaerolineaceae* and decreased *Fusarium* and *Acremonium* [49]. *Caldilineaceae*, the thermophilic bacteria, were increased by the addition of chicken manure to the soil [83], which accounted for 40–46% of *Chloroflexi* in arugula compost samples.

In contrast to arugula vermicompost, *Firmicutes* bacteria dominated in arugula compost [73]. These spore-forming bacteria can tolerate high temperatures during the composting process [84]. Most of them belong to *Bacillus* and *Clostridium* [85]. The genus *Bacillus* includes many species, most of which have been isolated from soil and have plant-promoting properties. They have the properties of nutrient solubilization, nitrogen fixation, production of siderophores, phytohormones, hydrogen cyanide (HCN) and lytic enzymes [86,87].

## Conclusions

Arugula (*Eruca sativa*) vermicompost was the most effective compost for suppressing *Meloidogyne javanica* and reducing its damage. The study of the quality and microbial structure of vermicompost showed the difference with arugula compost. On the other hand, vermicompost

had lower pH, EC and lower C/N ratio. The predominant bacterial phylum in vermicompost belongs to *Proteobacteria*. In addition, the combination of arugula vermicompost, bacteria (*Pseudomonas resinovorans* + *Sphingobacterium daejeonense*) and chitosan can act synergistically to limit the reproduction of *M. javanica* and improve plant growth. Due to the presence of chitinolytic and detoxifying PGPR bacteria in vermicompost, this combination is an environmentally friendly approach to control root-knot nematode.

The combination of arugula vermicompost with bacteria and chitosan reduced nematode infestation and has the potential to be marketed and used in agriculture. Efforts were made to select the amount of components used in this composition to be effective and cost-efficient under field conditions. The nematicidal activity of the combination of arugula vermicompost + bacteria (*Pseudomonas resinovorans* + *Sphingobacterium daejeonense*) + chitosan and its mode of action need to be studied under field conditions. Further studies comparing compost and vermicomposts in terms of enzyme activity, fungal community, and plant growth hormones are strongly recommended.

## Supporting information

**S1 Fig. Electrophoresis pattern of V3-V4 region of 16S rDNA gene of bacteria in compost and vermicompost of arugula (M: Ladder; neg: Negative control (water); 1 & 2: Arugula compost samples; 3 & 4: Arugula vermicompost).**
(DOCX)

**S2 Fig. The different abundance of bacterial community differential heat map of vermicompost and compost of arugula.** Euclidean distance as a similarity metric and complete linkage method. (M: Arugula compost; VM: Arugula vermicompost).
(DOCX)

**S1 Table. Sequences of the primer sets used for amplifying the V3-V4 region of the 16S rDNA gene of bacteria.**
(DOCX)

**S2 Table. Thermal program for amplifying V3-V4 region of 16S rDNA gene of bacteria.**
(DOCX)

**S1 Raw images.**
(PDF)

## Author Contributions

**Conceptualization:** Mahsa Rostami, Akbar Karegar.

**Data curation:** Mahsa Rostami.

**Formal analysis:** Mahsa Rostami, Akbar Karegar, Abozar Ghorbani.

**Funding acquisition:** Akbar Karegar.

**Investigation:** Mahsa Rostami.

**Methodology:** Mahsa Rostami.

**Project administration:** Akbar Karegar.

**Resources:** Akbar Karegar.

**Supervision:** Akbar Karegar.

**Writing – original draft:** Mahsa Rostami.

**Writing – review & editing:** Mahsa Rostami, Akbar Karegar, S. Mohsen Taghavi, Reza Ghasemi-Fasaei.

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
