## [Decision Letter · Decision Letter 0]

7 Mar 2023

PONE-D-22-34501Effective combination of arugula vermicompost, chitin and inhibitory bacteria for suppression of the root-knot nematode Meloidogyne javanica and explanation of their beneficial properties based on microbial analysisPLOS ONE

Dear Dr. Karegar,

Thank you for submitting your manuscript to PLOS ONE. After careful consideration, we feel that it has merit but does not fully meet PLOS ONE’s publication criteria as it currently stands. Therefore, we invite you to submit a revised version of the manuscript that addresses the points raised during the review process.

We look forward to receiving your revised manuscript.

Kind regards,

Durgesh Kumar Jaiswal, Ph.D.

Academic Editor

PLOS ONE

Journal Requirements:

"The authors gratefully acknowledge the financial support from Shiraz University."

We note that you have provided additional information within the Acknowledgements Section. Please note that funding information should not appear in the Acknowledgments section or other areas of your manuscript. We will only publish funding information present in the Funding Statement section of the online submission form.

"the financial support from Shiraz University."

 "the financial support from Shiraz University."  

"The authors declare no conflict of interest"

7. Please upload a new copy of Figures 2, 3, 4, 5 and 6 as the detail is not clear. Please follow the link for more information: https://blogs.plos.org/plos/2019/06/looking-good-tips-for-creating-your-plos-figures-graphics/" https://blogs.plos.org/plos/2019/06/looking-good-tips-for-creating-your-plos-figures-graphics/

8. PLOS ONE now requires that authors provide the original uncropped and unadjusted images underlying all blot or gel results reported in a submission’s figures or Supporting Information files. This policy and the journal’s other requirements for blot/gel reporting and figure preparation are described in detail at https://journals.plos.org/plosone/s/figures#loc-blot-and-gel-reporting-requirements and https://journals.plos.org/plosone/s/figures#loc-preparing-figures-from-image-files. When you submit your revised manuscript, please ensure that your figures adhere fully to these guidelines and provide the original underlying images for all blot or gel data reported in your submission. See the following link for instructions on providing the original image data: https://journals.plos.org/plosone/s/figures#loc-original-images-for-blots-and-gels.   

Additional Editor Comments:

Dear co-authors,

Thank you for submitted your manuscript in our journal. However, after carefully reviewing the your manuscript, I have found the article required the major revision on behalf of reviewers comments.

Therefore, please revised the your manuscript by following reviewer comments and submit the revised manuscript with reviewer response.

Thank You

Reviewers' comments:

Reviewer's Responses to Questions

**Comments to the Author**

1. Is the manuscript technically sound, and do the data support the conclusions?

Reviewer #1: Partly

Reviewer #2: Yes

2. Has the statistical analysis been performed appropriately and rigorously? 

Reviewer #1: I Don't Know

Reviewer #2: Yes

3. Have the authors made all data underlying the findings in their manuscript fully available?

Reviewer #1: Yes

Reviewer #2: Yes

4. Is the manuscript presented in an intelligible fashion and written in standard English?

Reviewer #1: No

Reviewer #2: Yes

5. Review Comments to the Author

Reviewer #1: 1. A revision of english by a mother-language expert is needed throughout the manuscript.

2. Materials and Methods should be also more concise, avoiding to describe the treatments of each experiment, as reported in Table 1.

3. The protocols of the second, third and fouth experiments are incomplete. You should have included also treatments with bacteria alone, both single or in combination, as to assess which is their contribution to plant growth and nematicidal effects. You stated that their effects were stated in a previous studies, but it was necessary to verify these effects also in the specific conditions of these experiments. For the same considerations, the combination of chitosan with bacterial agents should have been included in the third and fourth experiment.

4. In the Tables, there are too many and often redundant plant growth and nematode parameters. It could have been better to show only the results of the most representative parameters.

5. The Discussion section should be rearranged more fluently and not as a sequence of short sentences sometimes difficult to connect and understand. Long lists of bacterial families should be avoided in the text and transferred in tables.

6. In the Discussion you should also clarify if the bacterial species used in these experiments could have a potential development on an industrial scale at reasonable costs for their application in field.

7. You should revise also the reference section, removing all those not strictly necessary and including those missing.

8. Line 27: nematodes are parasites, not pathogens.

9. Line 31: castor.

10. Line 34-36: Specify that it is the extract from these material

11. Line 36: specify “Soil amendments with …”

12. Lines 36-41: the two sentences state almost the sale result, so remove the first sentence or specify better its difference from the following.

13. Line 56: vermicompost can be better defined as a soil amendment, biocontrol agents are fungi and bacteria.

14. Line 65-67: add literature references to this statement.

15. Lines 74-75: add refferences.

16. Lines 95-96: remove the sencence “there are no Studies …”, as repeated also in the following paragraph.

17. Line 109: the nematode was reared.

18. Line 110: describe this technique more in detail and add a related reference

19. Line 144: how did you choice that extract concentration? Explain.

20. Line 146: did you check if the juveniles were really dead and not only immobilized? If yes, how did you check it? Moreover, why did you stop the egg hatch after 72 hour, as it coul continue also later? Did you verify this?

21. Line 154: there is no mention of sulfur in the introduction as to justify its presence in this experiment

22. Lines 182-184: not necessary to list the treatments, are reported in table 1.

23. Line 184: did you use raw or composted animal manure? in analogy wth the comparison between vermicompost and compost of the other material, you should have been used vermicompost and compost from animal manure. in analogy wth the comparison between vermicompost and compost of the other material, you should have been used vermicompost and compost from animal manure.

24. Line 189: the extraction of eggs in NaOCl heavily reduce the egg hatchability, so this egg inoculum could have been a much lower dendity of viable eggs.

25. Line 195: describe better this tray method and add references.

26. Line 200: was it different from the commercial chitosan used in the first experimen? How did you chice the dose?

27. Lines 202-203: How these bacterial inocula were prepared? You should specify before describing the experiments.Moreover, it is not necessary to list the bacterial specie, as reported in Table 1.

28. Lines 208-211: as above, treatments have been already described in Table 1, so you can remove tis sentence.

29. Lines 269-270: this title is tt detailed, shorten it.

30. Line 273: the extracts from composts and vermicoposts…

31. Line 279-280: shorten this title

32. Line 287: you should explain in the Discussion why there were significant effects on dry weigh a nd not on the fresh weight.

33. Lines 289-293: You should explain in the Discussion how the nematode population in soil treated with the arugula vermicompost decreased while the nematode multiplication on tomato roots was not significantly affected.

34. Line 330, 318 and 350: titles should be in agreement with those of previous experiments.

35. Line 321-324: you state before that all treatmrnts increased shoot dry weight and then that some of them were not significantly different from the control. Check better this sentence.

36. Line 381: the bacterial 16S rRNA gene from compost and vermicompost…

37. Lines 404-418 and lines 419-425: these families can be listed, with their % presence, in a separate table and not in the text.

38. Lines 426-431: this list of phyla with % presence can be showed in a table and not in the text.

39. Lines 445-446: this statement is true only for the second experiment, specify.

40. Line 457-458: rewrite this sentence

41. Line 459: What is granular sulfur compost? Explain better.

42. Line 473: nematode penetration, not inoculation.

43. Line 473: “Chitosan contains…” add a reference to statements of this sentence.

44. Lines 475-477: rewrite this sentence, it is not clear.

45. Lines 480-481: “This plants belongs….”, add a reference.

46. Lines 481-482: add a reference.

47. Line 540: the genus is Devosia.

Reviewer #2: This manuscript entitled as 'Effective combination of arugula vermicompost, chitin and inhibitory bacteria for suppression of the root-knot nematode Meloidogyne javanica and explanation of their beneficial properties based on microbial analysis' deals with organic amendments and bacteria on plant-parasitic nematode as well as their combination.

I have some comments on the study:

1. Why there `are no commercial product available used to compare your treatments results?

2. Is it possible to commercialize your treatments? what is the obstacles to do so? cost? time?

3. why you did not do recovery test for nematodes under the laboratory conditions?

4. why root weight with no nematodes was not studied? You did for shoot weight only!

5. Table 1 - should stand alone and in the current form not easy for audience to understand.

6. English must be considered.

6. PLOS authors have the option to publish the peer review history of their article (what does this mean?). If published, this will include your full peer review and any attached files.

Reviewer #1: No

Reviewer #2: No

---

## [Author Response · Author response to Decision Letter 0]

25 Apr 2023

April 20, 2023

Prof. Durgesh Kumar Jaiswal

Editor in Chief

PLOS ONE

Dear Prof. Durgesh Kumar Jaiswal,

First, my co-authors and I would like to thank you and the two anonymous reviewers for considering our manuscript PONE-D-22-34501 with a positive attitude. We are very grateful to the reviewers for their comments, which clearly helped us to improve the quality of this manuscript. We agree with the reviewers on many points and have listed our response point by point at the end of this letter to explain why.

We hope that both you and the reviewers appreciate our revised text, and we are always available if you need further clarification on the revised text. For your information, all changes in the text are marked as "Highlight Text."

We thank you again for your time and attention and look forward to making a final decision on the manuscript.

Kind regards

Akbar Karegar

Reviewers' Comments to Author:

Reviewer: 1

1. A revision of english by a mother-language expert is needed throughout the manuscript.

- We have not found a colleague who is proficient in English writing. But all authors have revised the manuscript to check its spelling and English structure. We have improved the text as much as possible. The changes are highlighted in the text.

2. Materials and Methods should be also more concise, avoiding to describe the treatments of each experiment, as reported in Table 1.

- "Table 1" has been deleted and the number of other tables in the main text has been corrected. This sentence was deleted "The treatments of all experiments are shown in Table 1".

3. The protocols of the second, third and fouth experiments are incomplete. You should have included also treatments with bacteria alone, both single or in combination, as to assess which is their contribution to plant growth and nematicidal effects. You stated that their effects were stated in a previous studies, but it was necessary to verify these effects also in the specific conditions of these experiments. For the same considerations, the combination of chitosan with bacterial agents should have been included in the third and fourth experiment.

- Since the conditions and method of obtaining the results are similar in the greenhouse experiments, we refrain from mentioning them again and only explain the changes in the experiment.

The effect of the bacteria mentioned in the manuscript on root-knot nematode was demonstrated in our previous study (https://doi.org/10.1186/s41938-021-00383-9), following the principles of reproducibility of the results. In the current study, we tried to find the best combination for effective nematode control and plant growth improvement in several steps of the greenhouse experiments.

Therefore, in each step of the trials, the treatments with the best results were selected and combined with new treatments in the next step. If the results of the combination were effective, it was included in the next step of the experiment. For this reason, the treatments of the third phase of the greenhouse experiment were repeated exactly in the fourth phase.

Since the combination of bacteria with different grams is not effective, it was preferred to use only the combination of bacteria of the same gram.

In order to obtain accurate results and control the conditions in the greenhouse, the test method and the number of treatments and repetitions were planned in this way. 

4. In the Tables, there are too many and often redundant plant growth and nematode parameters. It could have been better to show only the results of the most representative parameters.

The column of ‘J2s / pot soil’ in Tables 1, 2, 3 and 4, column ‘Gall / root’ in Table 3 and column ‘Root fresh weigh’ in Table 4 were deleted.

Important and significant parameters are mentioned in the tables.

5. The Discussion section should be rearranged more fluently and not as a sequence of short sentences sometimes difficult to connect and understand. Long lists of bacterial families should be avoided in the text and transferred in tables.

- The discussion section has been revised and all the changes in the text have been highlighted. 

The list of bacteria in the text transferred in Table 5 

6. In the Discussion you should also clarify if the bacterial species used in these experiments could have a potential development on an industrial scale at reasonable costs for their application in field.

- This matter was added and explained in the discussion section. The references were added to the reference section of the manuscript (42).

7. You should revise also the reference section, removing all those not strictly necessary and including those missing.

- The reference section was revised. (2, 5, 6, 11, 13, 16, 20, 43, 46, 47, 52, 62, 74, 84, and 87 references were deleted. The numbers mentioned are in the first version)

8. Line 27: nematodes are parasites, not pathogens.

- Pathogens was changed to the parasites 

9. Line 31: castor.

-It was changed to the castor 

10. Line 34-36: Specify that it is the extract from these material

-The words "The extract of" were added. 

11. Line 36: specify “Soil amendments with …”

-It was done 

12. Lines 36-41: the two sentences state almost the sale result, so remove the first sentence or specify better its difference from the following.

-It was deleted 

13. Line 56: vermicompost can be better defined as a soil amendment, biocontrol agents are fungi and bacteria.

-It was done 

14. Line 65-67: add literature references to this statement.

- These statements have the reference 6 (https://doi.org/10.1186/2193-1801-1-26)

15. Lines 74-75: add references.

-the "9" and "10" references were added 

16. Lines 95-96: remove the sencence “there are no Studies …”, as repeated also in the following paragraph.

-It was deleted 

17. Line 109: the nematode was reared.

-It was done 

18. Line 110: describe this technique more in detail and add a related reference

-the sentences were added and the related references also added to the references section (21, 22) 

19. Line 144: how did you choice that extract concentration? Explain.

All compost and vermicompost extracts were extracted uniformly.

 For the mixtures of nematode eggs-water and larva-water, the number of nematodes was calibrated so that there were 100 nematodes to one milliliter of water. Then the extract was added to the Petri dishes in an amount that covered the surface of the Petri dish; by rotating the Petri dish, the mixture of eggs and extract was evenly distributed in it and could be counted.

This method was used in our previous project (https://doi.org/10.1186/s41938-021-00383-9 and https://doi.org/10.1007/s40093-014-0058-y ). In the current study, the best method for laboratory testing was tried so that an even mixture of eggs, larvae and extracts was obtained and their counting was easily possible under binoculars.

20. Line 146: did you check if the juveniles were really dead and not only immobilized? If yes, how did you check it? Moreover, why did you stop the egg hatch after 72 hour, as it coul continue also later? Did you verify this?

To control the juveniles, a special needle was used, which was struck during the count, and when they were completely motionless and dead, they were counted. Each Petri dish was counted three times to reduce the possible error.

The uniformity and regularity of treatments and replicates may decrease after 72 hours. This is because after this time, the test error increases and the sterility of the environment may also decrease. It is possible that factors other than those studied may also affect the test.

These times have been checked in previous projects and the best time was selected for the current test (https://doi.org/10.1186/s41938-021-00383-9 and https://doi.org/10.1007/s40093-014-0058-y ).

21. Line 154: there is no mention of sulfur in the introduction as to justify its presence in this experiment

One of the interesting properties of the arugula plant is the presence of sulfur derivatives, which is also mentioned in the discussion section.

Sulfur was included in the treatments for comparison purposes, but the results showed that the effect of arugula vermicompost and compost was greater than the use of sulfur. In the text, the effect of arugula was stated in competition with sulfur, and since it is one of the auxiliary components of the study to determine the effect of the main components, it is not explained in the introduction, but it is mentioned in the discussion.

22. Lines 182-184: not necessary to list the treatments, are reported in table 1.

-they were deleted 

23. Line 184: did you use raw or composted animal manure? in analogy wth the comparison between vermicompost and compost of the other material, you should have been used vermicompost and compost from animal manure. in analogy wth the comparison between vermicompost and compost of the other material, you should have been used vermicompost and compost from animal manure.

"Animal manure" in the treatments is composted animal manure, and "common vermicompost" is the vermicompost of animal manure.

Ordinary vermicompost is usually made from animal manure that is why we called it so.

Raw animal manure is harmful to the plant, so it is usually used before planting the plants in the field or greenhouse. Since the animal manure was used at the same time as planting, it is composted.

Both treatments were explained in the material method section 

24. Line 189: the extraction of eggs in NaOCl heavily reduce the egg hatchability, so this egg inoculum could have been a much lower dendity of viable eggs.

NaClO is used in low concentration. In this way, gelatin is removed from the egg mass and eggs can be counted separately. In addition to the controlled NaClO concentration, the eggs are washed with sterile water after extraction to remove residual NaClO.

Furthermore, the general conditions and environmental conditions for the treatments were identical to minimize experimental errors.

25. Line 195: describe better this tray method and add references.

- explain of this method were added. The related reference was added to the reference section of the manuscript (27).

26. Line 200: was it different from the commercial chitosan used in the first experimen? How did you chice the dose?

Commercial chitosans that can be used as fertilizers are usually liquid and acid soluble. Acid can degrade this quality. Therefore, it was decided to extract chitosan from shrimp shell and use it in the study. Pure chitosan was used for the experiments. We also confirmed the quality of the chitosan with an FTIR test.

The addition of powdered chitosan to vermicompost is a new method. Compared to the commercial chitosans, most of which are acid soluble, and also according to the same article where chitin was added to vermicompost (0.004%) (Castro, L., Flores, L., & Uribe, L. (2011). Effect of vermicompost and chitin on the control of Meloidogyne incognita in greenhouse tomato. Agronomía Costarricense, 35(2), 21-32.) and also according to the higher price and quality of chitosan compared to our chitin. For the greenhouse test, we used 0.001%. This cost percentage is more justifiable. This material has a very low density, so the volume of its use is significant compared to the specified percentage. 

FTIR test

27. Lines 202-203: How these bacterial inocula were prepared? You should specify before describing the experiments.Moreover, it is not necessary to list the bacterial specie, as reported in Table 1.

Preparation of bacteria for inoculation was added to the material and method. The names of the bacteria were deleted from the second and third experiment sections of the material and method.

28. Lines 208-211: as above, treatments have been already described in Table 1, so you can remove tis sentence.

- They were deleted 

29. Lines 269-270: this title is tt detailed, shorten it.

-It was done 

30. Line 273: the extracts from composts and vermicoposts…

- "the extracts from" was added 

31. Line 279-280: shorten this title

-It was done 

32. Line 287: you should explain in the Discussion why there were significant effects on dry weigh a nd not on the fresh weight.

- This matter was added to the discussion part 

33. Lines 289-293: You should explain in the Discussion how the nematode population in soil treated with the arugula vermicompost decreased while the nematode multiplication on tomato roots was not significantly affected.

- This matter was added to the discussion part 

34. Line 330, 318 and 350: titles should be in agreement with those of previous experiments.

-The name of glasshouse experiments were completed and changed in the result section.

35. Line 321-324: you state before that all treatmrnts increased shoot dry weight and then that some of them were not significantly different from the control. Check better this sentence.

The sentences were changed to correct this matter 

36. Line 381: the bacterial 16S rRNA gene from compost and vermicompost…

-It was done 

37. Lines 404-418 and lines 419-425: these families can be listed, with their % presence, in a separate table and not in the text.

They were listed in "Table 5"

38. Lines 426-431: this list of phyla with % presence can be showed in a table and not in the text.

 The presence of three phyla with related order was listed in table 5, but some information about number of OUT and name of phyla are in the text.

39. Lines 445-446: this statement is true only for the second experiment, specify.

These sentences are about the first experiment. They were changed and completed 

40. Line 457-458: rewrite this sentence

-It was done 

41. Line 459: What is granular sulfur compost? Explain better.

The explanation was added 

42. Line 473: nematode penetration, not inoculation.

-It was done 

43. Line 473: “Chitosan contains…” add a reference to statements of this sentence.

The reference was add (38)

44. Lines 475-477: rewrite this sentence, it is not clear.

-It was done 

45. Lines 480-481: “This plants belongs….”, add a reference.

The reference was add (40) 

46. Lines 481-482: add a reference.

The reference "32" is related to these sentences. This reference is a review article and we used it. (https://doi.org/10.1016/j.apsoil.2009.11.003)

47. Line 540: the genus is Devosia.

The genus was changed to the species 

Reviewer: 2

1. Why there `are no commercial product available used to compare your treatments results?

Since the objective of this project is to obtain a combination of organic materials to control nematode damage, attempts were made to apply the appropriate treatments.

In the first greenhouse experiment, we used vermicompost and animal manure compost, which are normally used in agriculture, in relation to compost and vermicompost of inhibitor plants. The effect of animal manure vermicompost has been demonstrated in many articles, and the aim of this study is to improve the results of previous studies.

In addition, commercial chitosan was used to compare the treatment with shrimp shells.

2. Is it possible to commercialize your treatments? what is the obstacles to do so? cost? time?

This matter was added to the manuscript. 

3. why you did not do recovery test for nematodes under the laboratory conditions?

Under laboratory conditions, the effect of compost extracts on nematode eggs and larvae is studied directly, while under greenhouse conditions this effect is different and the effect of composts is indirect. Therefore, the laboratory tests cannot show the real effect on the plant. Our laboratory studies were a preliminary evaluation for greenhouse studies. Since greenhouse results simulate reality more reliably, they were repeated in several steps to determine the effective combination for controlling nematode damage.

4. why root weight with no nematodes was not studied? You did for shoot weight only!

Because of the galls present in the roots, root weight is not a reliable measure for comparing healthy and infected plants.

The fresh weight of the roots of infected plants is used to calculate the number of eggs, egg mass, and galls in one gram of the root.

The most important parameter for comparing infected healthy plants in this study is the fresh weight and dry weight of plant shoots.

5. Table 1 - should stand alone and in the current form not easy for audience to understand.

According to the comment of the first reviewer, Table 1 was removed from the manuscript.

6. English must be considered.

- We did not found colleague proficient in English writing. But all authors have revised the manuscript to check its writing and English structure. We have improved the text as much as possible. The changes are highlighted in the text.

---

## [Decision Letter · Decision Letter 1]

17 Jul 2023

PONE-D-22-34501R1Effective combination of arugula vermicompost, chitin and inhibitory bacteria for suppression of the root-knot nematode Meloidogyne javanica and explanation of their beneficial properties based on microbial analysisPLOS ONE

Dear Dr. Karegar,

Thank you for submitting your manuscript to PLOS ONE. After careful consideration, we feel that it has merit but does not fully meet PLOS ONE’s publication criteria as it currently stands. Therefore, we invite you to submit a revised version of the manuscript that addresses the points raised during the review process.

We look forward to receiving your revised manuscript.

Kind regards,

RAJA AADIL HUSSAIN BHAT

Academic Editor

PLOS ONE

Journal Requirements:

Additional Editor Comments :

The authors are advised to incorporate the suggestions of reviewers.

Reviewers' comments:

Reviewer's Responses to Questions

**Comments to the Author**

1. If the authors have adequately addressed your comments raised in a previous round of review and you feel that this manuscript is now acceptable for publication, you may indicate that here to bypass the “Comments to the Author” section, enter your conflict of interest statement in the “Confidential to Editor” section, and submit your "Accept" recommendation.

Reviewer #2: All comments have been addressed

Reviewer #3: (No Response)

2. Is the manuscript technically sound, and do the data support the conclusions?

Reviewer #2: Yes

Reviewer #3: Yes

3. Has the statistical analysis been performed appropriately and rigorously? 

Reviewer #2: Yes

Reviewer #3: Yes

4. Have the authors made all data underlying the findings in their manuscript fully available?

Reviewer #2: Yes

Reviewer #3: Yes

5. Is the manuscript presented in an intelligible fashion and written in standard English?

Reviewer #2: Yes

Reviewer #3: No

6. Review Comments to the Author

Reviewer #2: The authors have improved the manuscript substantially. I am sure the editorial board will go through and fix the English before sending the proof to the authors.

Reviewer #3: (No Response)

7. PLOS authors have the option to publish the peer review history of their article (what does this mean?). If published, this will include your full peer review and any attached files.

Reviewer #2: No

Reviewer #3: No

---

## [Author Response · Author response to Decision Letter 1]

27 Jul 2023

July 27, 2023

Prof. Raja Aadil Hussain Bhat

Editor in Chief

PLOS ONE

Dear Prof. Raja Aadil Hussain Bhat,

First, my co-authors and I would like to thank you and the two anonymous reviewers for considering our manuscript PONE-D-22-34501 with a positive attitude. We are very grateful to the reviewers for their comments, which clearly helped us to improve the quality of this manuscript. We agree with the reviewers on many points and have listed our response point by point at the end of this letter to explain why.

We hope that both you and the reviewers appreciate our revised text, and we are always available if you need further clarification on the revised text. For your information, all changes in the text are marked as "Highlight Text." Below are the reviewers' comments, and our responses (RED text). Also, we check the adjustment of figures manuscript by the Preflight Analysis and Conversion Engine (PACE) that figures met PLOS requirements. We reviewed references and they also were checked by "http://retractiondatabase.org".

We thank you again for your time and attention and look forward to making a final decision on the manuscript.

Kind regards

Major Comments:

The introduction should be restructured to clearly outline the purpose of the research and its significance. It should highlight the problem of nematode infestation in agriculture and the need for eco-friendly and sustainable solutions. 

A comprehensive literature review should be conducted to explore existing research on the use of vermicompost as a biocontrol agent against nematodes. Cite relevant studies that have demonstrated the efficacy of vermicompost in nematode control. Discuss the limitations and gaps in the current knowledge that this research aims to address.

In response to your suggestion, we have restructured the introduction to more explicitly outline the purpose of our research and emphasize its significance in addressing the problem of nematode infestation in agriculture. The revised introduction now provides a more comprehensive overview of the importance of eco-friendly and sustainable solutions for managing nematode pests and the utilization of vermicompost as a biocontrol agent based on existing research findings. The relevant references were added to the manuscript.

2: Pajovic, I., Širca, S., Stare, B. G., & Urek, G. (2007). The incidence of root-knot nematodes Meloidogyne arenaria, M. incognita, and M. javanica on vegetables and weeds in Montenegro. Plant disease, 91(11), 1514-1514.

3: Pathak, V. M., Verma, V. K., Rawat, B. S., Kaur, B., Babu, N., Sharma, A., ... & Cunill, J. M. (2022). Current status of pesticide effects on environment, human health and it’s eco-friendly management as bioremediation: A comprehensive review. Frontiers in Microbiology, 2833.

4: Forghani, F., & Hajihassani, A. (2020). Recent advances in the development of environmentally benign treatments to control root-knot nematodes. Frontiers in plant science, 11, 1125.

8: Mondal, S., Ghosh, S., & Mukherjee, A. (2021). Application of biochar and vermicompost against the rice root-knot nematode (Meloidogyne graminicola): an eco-friendly approach in nematode management. Journal of Plant Diseases and Protection, 128, 819-829.

9: Liu, D., Han, W., Zhang, Y., & Jiang, Y. (2019). Evaluation of vermicompost and extracts on tomato root-knot nematode. Bangladesh Journal of Botany, 48(3 Special), 845-851.

10: Zuhair, R., Moustafa, Y. T. A., Mustafa, N. S., El-Dahshouri, M. F., Zhang, L., & Ageba, M. F. (2022). Efficacy of amended vermicompost for bio-control of root knot nematode (RKN) Meloidogyne incognita infesting tomato in Egypt. Environmental Technology & Innovation, 27, 102397.

11: Safiddine, F., Nebih, D., Merah, O., & Djazouli, Z. E. (2019). The impact of different vermicompost types on reducing the number of Meloidogyne root-knot nematodes and the vegetative expression of tomato plants. AgroBiologia, 9(1), 1415-1427.

Minor Comments:

1) "The earthworms could not feed on the castor bean plant, and its vermicompost was not produced." - Justification for this statement should be included, explaining why earthworms couldn't feed on the castor bean plant and how it impacted vermicompost production.

This was done and the relevant reference was added to the manuscript.

The fresh leaves of the plant cannot be utilized as earthworm feed due to their tendency to become soggy, decay, and release harmful exudates that are lethal to earthworms. 

37. Antony Godson SG, Gajalakshmi S. exploring the possibility of utilizing castor bean (Ricinus communis) plant leaves in vermicomposting with three epigeic earthworm species. J Emerg Technol Innov Res.2018; 12(5): 501-507.

38. Mahadev ND, Thorat AT, Vitthal BP. An evaluation of anthelmintic activity of Ricinus communis Linn. leaves by using different type of solvent. J Pharmacogn Phytochem. 2017;6(4):1845-7.

2) "The predominant bacterial phylum in vermicompost belongs to a gram-negative group" - Mention the specific name of the gram-negative group.

This was done 

The predominant bacterial phylum in vermicompost belongs to Proteobacteria

3) "This study was conducted with to introduce an effective combination to control the damage caused by the root-knot nematode." - Correct the sentence for clarity, possibly: "This study aims to introduce an effective combination for controlling root-knot nematode damage."

It was corrected in the text.

4) English language and writing should be improved, possibly by seeking the assistance of a native speaker for proofreading and editing. This will help ensure clarity and proper communication of the research findings.

All authors have revised the manuscript to check its spelling and English structure. We have improved the text as much as possible. The changes are highlighted in the text.

---

## [Editor Report · Decision Letter 2]

31 Jul 2023

Effective combination of arugula vermicompost, chitin and inhibitory bacteria for suppression of the root-knot nematode Meloidogyne javanica and explanation of their beneficial properties based on microbial analysis

PONE-D-22-34501R2

Dear Dr. Karegar,

We’re pleased to inform you that your manuscript has been judged scientifically suitable for publication and will be formally accepted for publication once it meets all outstanding technical requirements.

Kind regards,

RAJA AADIL HUSSAIN BHAT

Academic Editor

PLOS ONE
---

## [Editor Report · Acceptance letter]

3 Aug 2023

PONE-D-22-34501R2 

Effective combination of arugula vermicompost, chitin and inhibitory bacteria for suppression of the root-knot nematode *Meloidogyne javanica* and explanation of their beneficial properties based on microbial analysis 

Dear Dr. Karegar:

I'm pleased to inform you that your manuscript has been deemed suitable for publication in PLOS ONE. Congratulations! Your manuscript is now with our production department. 

Kind regards, 

on behalf of

Dr. RAJA AADIL HUSSAIN BHAT 

Academic Editor

PLOS ONE